# Mechanical force induces mitochondrial fission

Sebastian Carsten Johannes Helle[1†], Qian Feng[1†], Mathias J Aebersold[2], Luca Hirt[2], Raphael R Grüter[2], Afshin Vahid[3], Andrea Sirianni[4], Serge Mostowy[4], Jess G Snedeker[5,6], Anđela Šarić[7], Timon Idema[3], Tomaso Zambelli[2], Benoît Kornmann[1]*

[1]Institute of Biochemistry, ETH Zurich, Zurich, Switzerland; [2]Laboratory of Biosensors and Bioelectronics, Institute for Biomedical Engineering, ETH Zurich, Zurich, Switzerland; [3]Department of Bionanoscience, Kavli Institute of Nanoscience, Delft University of Technology, Delft, Netherlands; [4]Section of Microbiology, MRC Centre for Molecular Bacteriology and Infection, Imperial College London, London, United Kingdom; [5]Balgrist University Hospital, University of Zurich, Zurich, Switzerland; [6]Institute for Biomechanics, ETH Zurich, Zurich, Switzerland; [7]Department of Physics and Astronomy, Institute for the Physics of Living Systems, University College London, London, United Kingdom

*For correspondence: benoit.kornmann@bc.biol.ethz.ch

†These authors contributed equally to this work

Competing interests: The authors declare that no competing interests exist.

**Abstract** Eukaryotic cells are densely packed with macromolecular complexes and intertwining organelles, continually transported and reshaped. Intriguingly, organelles avoid clashing and entangling with each other in such limited space. Mitochondria form extensive networks constantly remodeled by fission and fusion. Here, we show that mitochondrial fission is triggered by mechanical forces. Mechano-stimulation of mitochondria – via encounter with motile intracellular pathogens, via external pressure applied by an atomic force microscope, or via cell migration across uneven microsurfaces – results in the recruitment of the mitochondrial fission machinery, and subsequent division. We propose that MFF, owing to affinity for narrow mitochondria, acts as a membrane-bound force sensor to recruit the fission machinery to mechanically strained sites. Thus, mitochondria adapt to the environment by sensing and responding to biomechanical cues. Our findings that mechanical triggers can be coupled to biochemical responses in membrane dynamics may explain how organelles orderly cohabit in the crowded cytoplasm.
DOI: https://doi.org/10.7554/eLife.30292.001

## Introduction

Eukaryotic cells are densely packed with macromolecular complexes and intertwining membranous organelles (*Marsh et al., 2001*). Some organelles, such as the endoplasmic reticulum (ER) and mitochondria, assemble into highly branched and dynamic networks, adding complexity to the subcellular architecture. Moreover, cells constantly remodel their cytoplasm. For instance, vesicular and membrane trafficking continuously move large complexes and organelles across long distances in the cytoplasm. Given the limited space, it is surprising that large organelle networks can coexist without entanglement or encroachment. In the particular case of mitochondria, collisions and entanglements could lead to catastrophic consequences, such as leakage of cytochrome-C into the cytosol and the induction of apoptosis.

We thus postulate here that cells must be equipped with mechanisms allowing mitochondria to resolve potential stresses resulting from collisions with other structures, and investigate how mitochondria cope with intracellular crowdedness.

## Results

### Effect of *Shigella flexneri* actin-based motility on mitochondria

We wondered how mitochondria cope with being hit by an intracellular fast-moving object. *Shigella flexneri* are pathogenic bacteria belonging to the *Enterobacteriaceae* family, and infection in humans causes diarrhea and severe inflammation in the gut. Upon entry into the cytoplasm of infected cells, a sub-population of the bacteria hijacks the actin cytoskeleton and stimulates its polymerization on the bacterial surface, forming so-called actin comet tails (*Ray et al., 2009*), allowing them to propel rapidly through the cytoplasm reaching speeds of up to 0.5 μm/s (*Gouin et al., 1999*). We infected U2OS or COS7 cells with virulent, fluorescently labelled *S. flexneri* and visualized mitochondria using mitochondria matrix-targeted BFP (mtBFP) (*Kanfer et al., 2015*). Using time-lapse microscopy, we observed that bacteria oftentimes collided with mitochondria, pushing the mitochondrial tubules aside, above or below (*Figure 1A*, *Video 1*). In some cases, collisions caused a visible reduction of the mitochondrial fluorescence, indicating that the matrix was constricted. In 60% of such cases, mitochondria underwent fission at the constricted site within one to five minutes (n = 23; *Figure 1B* and *Video 2*). By contrast, we observed that merely 4% of non-stimulated mitochondria underwent fission within a five-minute time span.

Mitochondrial fission and fusion are two opposing processes that regulate mitochondrial morphology and connectivity. Both processes are highly regulated and culminate with specific recruitment of dynamin-related GTPases, which catalyze mitochondrial fission and fusion (*van der Bliek et al., 2013*). The fission GTPase DRP1 (Dynamin-related protein 1) assembles as homomultimeric rings around mitochondria and uses the energy of GTP hydrolysis to squeeze mitochondria, causing fission (*Francy et al., 2015*).

To assess whether the collision-associated mitochondria division events involved the canonical fission machinery, we imaged bacterial movement in DRP1-depleted cells. Here and throughout this manuscript, we achieved DRP1 depletion by three different approaches: (1) treatment with DRP1–directed siRNA, (2) lentiviral transduction of DRP1-directed shRNA, and (3) CRISPR-mediated mutagenesis of exon 2 (DRP1$^{CRISPR}$). All conditions led to efficient reduction of DRP1 levels (*Figure 1—figure supplement 1A–C*) and caused mitochondria to hyperfuse in both mock-infected and *Shigella*-infected cells. Mitochondria from DRP1-depleted cells were strongly affected by bacterial movements. They were pushed and dragged, and sometimes visibly thinned by constriction. However, in contrast to wild-type cells, mitochondria from DRP1$^{CRISPR}$ cells recovered without undergoing fission in 100% of the cases (n = 50, *Figure 1C*, *Video 3*, p<10$^{-7}$ from a Fisher's exact test), and despite the strong reduction in matrix staining during mechanical stimulation of mitochondria, we could observe that mitochondria remained connected at all time. Similarly, DRP1 siRNA treatment completely abolished motile *Shigella*-induced fission in wild-type cells (n = 19, p<10$^{-4}$). These results indicate that DRP1 is necessary for bacteria-induced fission.

To further confirm that the division events we observed in wild-type cells were *bona fide* fission events, we transfected cells with mCherry-tagged DRP1 (*Friedman et al., 2011*) and observed the recruitment of the mitochondrial fission machinery to the division sites. As reported previously, in non-infected cells, fluorescent protein-tagged DRP1 exhibited mostly diffuse cytosolic signal with bright foci on mitochondria, most of which stably associated with mitochondria, while a subset marked fission sites.

Upon *Shigella* infection, we observed DRP1 foci formation at sites where motile bacteria had crossed a mitochondrial tubule. These sites subsequently underwent fission (*Figure 1D*, *Video 4*). There were also events where *Shigella* hit mitochondrial regions that were already marked by weak DRP1 signal, which, upon impact, developed into more intense puncta and subsequently led to fission (*Video 5*). Together with DRP1 depletion data, these results indicate that mitochondria react to collisions with bacteria by actively undergoing fission. The variability in the time elapsed between *Shigella* impact and eventual fission may reflect stochastic differences in DRP1 recruitment and activation kinetics.

### Mitochondrial fission induced using an Atomic Force Microscope

We wondered how the mitochondrial fission machinery could sense the presence of the bacterium. One possibility is that this detection is biochemical, through factors exposed on the bacterial

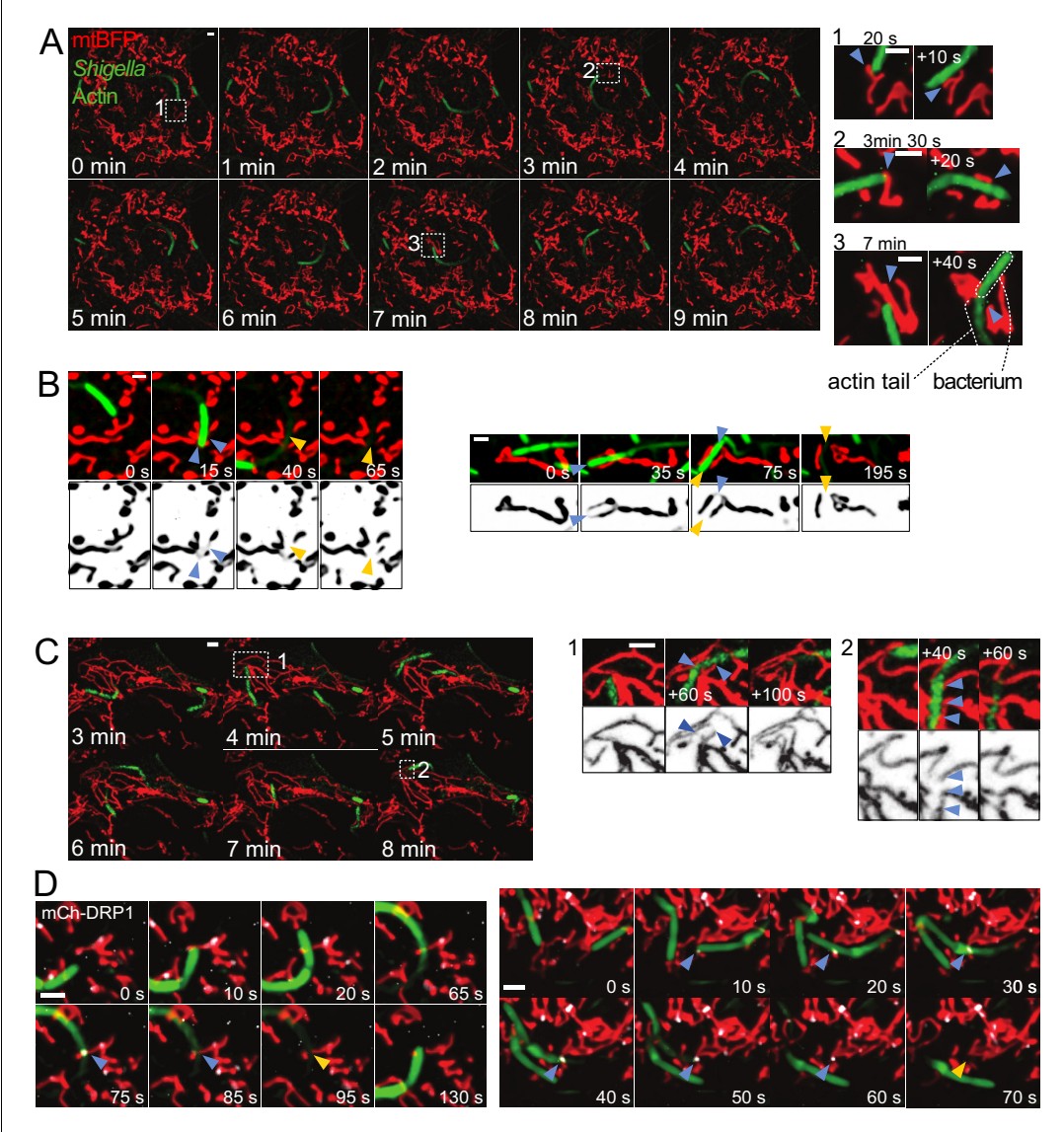

**Figure 1.** Mitochondria undergo DRP1-mediated fission upon encountering actin-propelled *Shigella*. (**A**) U2OS KERMIT cells (stably expressing mtBFP) were transfected with mCherry-Lifeact plasmid and infected with mCherry-labelled *S. flexneri*. Left, overview of time-lapse microscopy results presented at 1 min interval. Right, magnifications of selected areas from Left at the indicated times. Note that the times in the right inset do not necessarily match the times in the overview on the left. Arrowheads indicate mitochondria positions before and after impact with bacterium. (**B**) COS7 cells transduced with lentiviruses expressing GFP-Lifeact and mtBFP were infected with GFP-labelled *S. flexneri*. Imaging was performed as in A. Shown are four individual fission events upon encounter with *S. flexneri*. Blue and orange arrowheads indicate mitochondria before and after fission, respectively. (**C**) DRP1[CRISPR] U2OS KERMIT cells were subjected to the same treatment and analysis as in A. Numbered boxes as in A. Blue arrowheads, thinning mitochondrial tubules due to impact by *S. flexneri*, followed by recovery of mitochondrial tubules without fission. (**D**) mCherry-DRP1-expressing U2OS cells were treated as in B. Blue arrowheads, recruitment of DRP1 (white) at sites of encounter with *S. flexneri*. Orange arrowheads, subsequent fission events. Scale bars, 2 µm.

DOI: https://doi.org/10.7554/eLife.30292.002

The following figure supplement is available for figure 1:

**Figure supplement 1.** All detectable forms of DRP1 are depleted by siRNA, shRNA or CRISPR-induced mutations.

DOI: https://doi.org/10.7554/eLife.30292.003

surface. An alternative hypothesis is that mechanical forces imposed by the collision triggered mitochondrial fission.

In order to test more directly whether mechanical stimuli can cause mitochondrial fission, we employed atomic force microscopy (AFM) (*Binnig et al., 1986*). AFM can sense and/or transmit

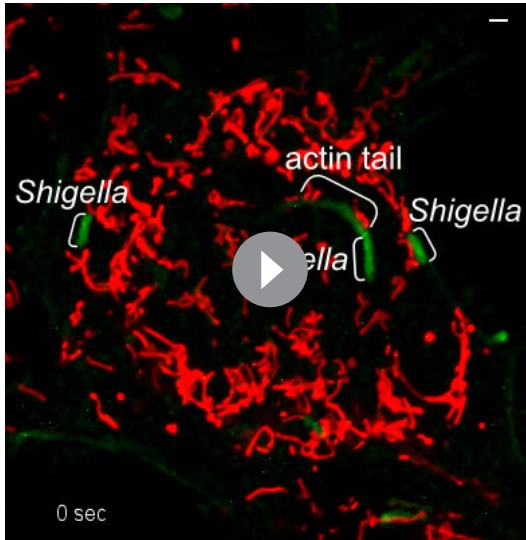

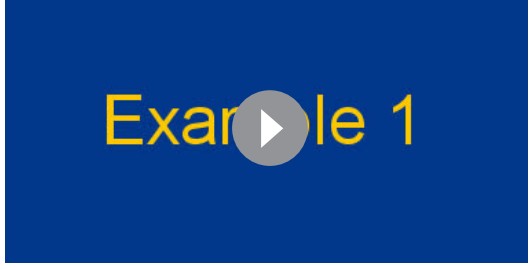

**Video 2.** Mitochondria divide upon encounter with *Shigella*. Cos7 cells transduced with lentiviruses expressing mtBFP and GFP-Lifeact were infected with GFP-labelled *S. flexneri*. Red, mitochondria. Green, *Shigella* and actin. Blue and orange arrowheads indicate mitochondria before and after *Shigella* - induced fission, respectively. Scale bar, 2 μm. This movie relates to *Figure 1B*.
DOI: https://doi.org/10.7554/eLife.30292.005

**Video 1.** Mitochondria are pushed aside upon impact with *Shigella*. U2OS KERMIT cells (stably expressing mtBFP) were transfected with mCherry-Lifeact plasmid and infected with mCherry-labelled *S. flexneri*. Red, mitochondria. Green, *Shigella* and actin. Arrowheads indicate events where mitochondrial tubules make way for *Shigella* upon encounter. Scale bar, 2 μm. This movie relates to *Figure 1A*.
DOI: https://doi.org/10.7554/eLife.30292.004

minute forces within the nanonewton (nN) range from and to the specimen, respectively, via a tip mounted on a cantilever (*Müller and Dufrêne, 2011*). Adherent cells have a very flat periphery (typically 100–200 nm thick [*Xu et al., 2012*]), hence we expected that applying pressure in these peripheral areas would induce deformation of the cell surface and transmit force to underlying mitochondria (*Figure 2A*).

Using a round AFM tip, we applied a defined force (set at 15 nN) to cells expressing an outer membrane marker – mCh-Fis1TM (mCherry targeted to the outer mitochondrial membrane by fusion to the transmembrane domain of Fis1) – and stained with Mitotracker Deep Red (targeting the matrix). AFM stimulus led to the constriction of mitochondria as evidenced by the reduction of both mitotracker and mCh-Fis1TM signals. Note that, for unknown reasons, the tip of the AFM was slightly fluorescent in the far red channel. Typically, mitochondria underwent fission at the tip contact point within one minute after the approach, and more rarely between 1 to 5 min (*Figure 2B–C*,

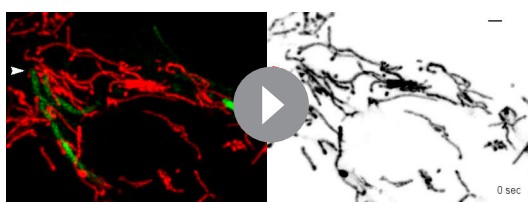

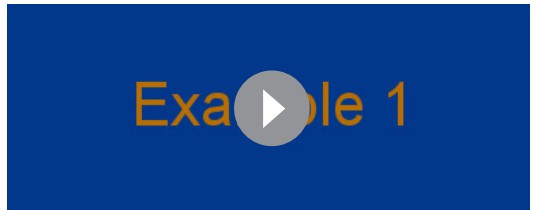

**Video 3.** Disturbance to mitochondrial morphology by *Shigella* in DRP1$^{CRISPR}$ knockout cells. DRP1$^{CRISPR}$ knockout U2OS KERMIT cells (stably expressing mtBFP) were transfected with mCherry-Lifeact plasmid and infected with mCherry-labelled *S. flexneri*. Red, mitochondria. Green, *Shigella* and actin. Arrowheads indicate thinning mitochondrial tubules due to impact by *Shigella*. One blue arrowhead (at 860 s) points to a DRP1-independent fission event that occurs away from the *Shigella* impact point. Scale bar, 2 μm. This movie relates to *Figure 1C*.
DOI: https://doi.org/10.7554/eLife.30292.006

**Video 4.** DRP1 recruitment and subsequent mitochondrial fission upon encounter with *Shigella*. U2OS cells transduced with lentiviruses expressing GFP-Lifeact and mtBFP were transfected with mCherry-DRP1 plasmid and infected with GFP-labelled *S. flexneri*. Red, mitochondria. Green, *Shigella* and actin. White, DRP1. Blue and orange arrowheads indicate mitochondria before and after *Shigella*-induced fission, respectively. Scale bar, 2 μm. This movie relates to *Figure 1D*.
DOI: https://doi.org/10.7554/eLife.30292.007

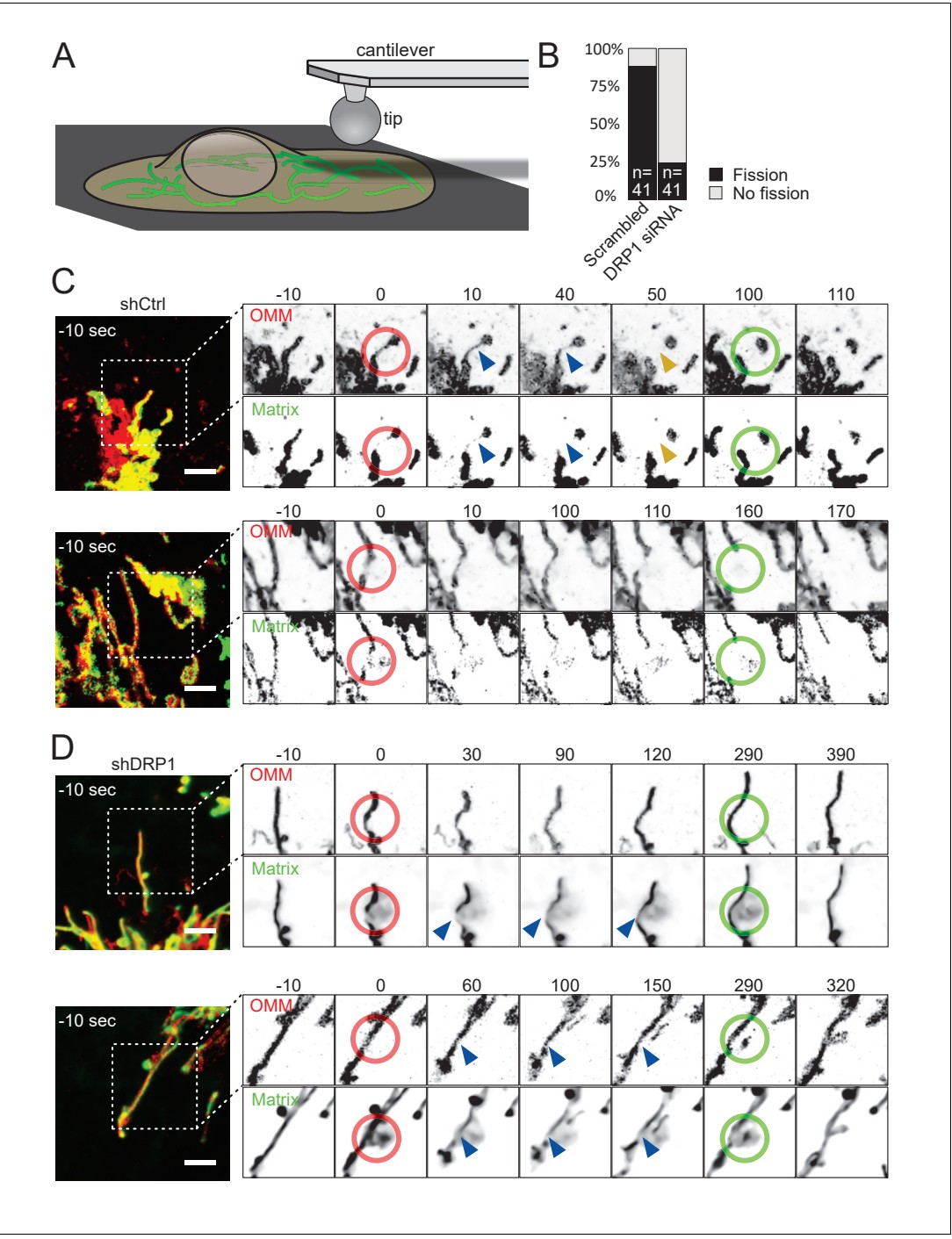

**Figure 2.** Mitochondria undergo DRP1-dependent fission upon AFM-mediated force application. (**A**) Scheme of the experimental setup. (**B**) Quantification of fission events elicited by the application of AFM-induced mechanical forces observed in cells treated either with scrambled or DRP1 siRNA. Successful force application to an individual mitochondrion was defined as visible constriction of the mitochondrial matrix following tip approach (examples shown with blue arrowhead in C and D). (**C**) U2OS cells stained with Mitotracker Deep Red and transduced with viruses encoding mCherry-FIS1TM and a control shRNA were imaged by time-lapse microscopy. Two examples are shown. At t = 0 s, the cantilever of the AFM approached the cell in Contact mode, with a force set at 15 nN, at the position of the red ring. Green rings mark the time and area of tip retraction. Blue arrowheads indicate mitochondria that are visibly thinned by the pressure but have not yet undergone fission. Fission events are indicated by an orange arrowhead. OMM panels, mCherry-FIS1TM (red). Matrix panel, mtBFP (green). Scale bar, 5 μm. (**D**) As in (**C**) except that the cells were treated with a virus encoding DRP1 shRNA. Scale bar, 5 μm.
*Figure 2 continued on next page*

*Figure 2 continued*

DOI: https://doi.org/10.7554/eLife.30292.009

The following figure supplement is available for figure 2:

**Figure supplement 1.** Mitochondria undergo DRP1-dependent fission upon AFM-mediated force application.
DOI: https://doi.org/10.7554/eLife.30292.010

*Video 6*). To ensure that mitochondria were actually divided, we retracted the AFM tip 50 s after the fission event occurred, and continued imaging the cells for another ~ 100 s. 36 of 41 successful approaches led to fission (success being scored as inducing a visible constriction of the mitochondrial matrix. All fission events were scored within 5 min).

As observed in the *Shigella* model, the delay between tip approach and mitochondrial fission was variable, likely reflecting the recruitment of the fission machinery. Indeed, these events were again DRP1-dependent, since repeating these experiments in DRP1-deficient cells led to a significant decrease in fission events upon force application (10 fission events in 41 attempts, $p<10^{-8}$; *Figure 2B,D*, *Video 7*). While mitochondria in DRP1-deficient cells were visibly constricted (*Figure 2D*, *Video 7*, blue arrows) upon and during 5 min of force application (red ring), they usually recovered from the force stimulus after tip retraction (green ring, *Figure 2B,D*, *Video 7*). We also acquired images using cells expressing a blue matrix marker (mtBFP) instead of mitotracker deep red, to avoid the fluorescence of the AFM tip, and similarly observed that AFM stimulation failed to induce mitochondrial fission in DRP1 shRNA-expressing cells, despite clear constrictions of the matrix (*Figure 2—figure supplement 1*, *Video 8*).

Although the force application can be precisely measured at the AFM tip, it is virtually impossible to know what fraction of the force was transduced to underlying mitochondria because the cortical actin meshwork likely buffered a large fraction. Of note, forces in the nanonewton range are typically needed to induce deformation of the actin cytoskeleton (*Hadjiantoniou et al., 2012*).

## Mitochondrial fission in cells grown on patterned substrates

External forces applied by an AFM tip might not be physiologically relevant for mitochondrial fission. We therefore sought an experimental system where cells themselves were responsible for force generation. We reasoned that morphological plasticity is an intrinsic property of many cell types that have to adapt to their natural microenvironment. Growing on or inside uneven and crowded tissues causes deformation of the cells, which may result in forces applied to the mitochondria therein. We therefore sought to mimic these conditions by culturing U2OS cells on uneven

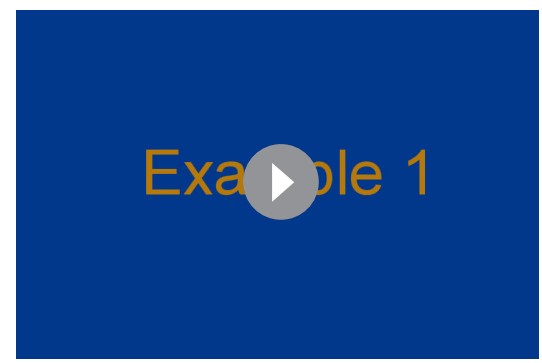

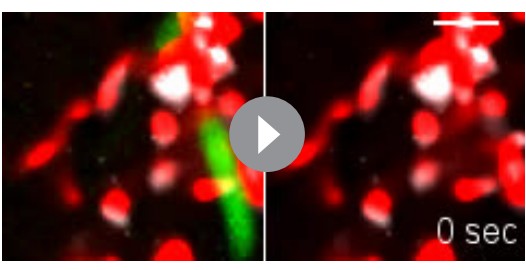

**Video 5.** DRP1 recruitment and subsequent mitochondrial fission upon encounter with *Shigella*. As in *Video 4*. Blue arrowheads indicate event where *Shigella* crosses a mitochondria region that was already coated with low level of DRP1. Orange arrowheads indicate formation of a bright DRP1 focus at this site, which subsequently undergoes fission. Scale bar, 2 μm.
DOI: https://doi.org/10.7554/eLife.30292.008

**Video 6.** Mitochondrial fission induced by contact with an AFM tip. U2OS cells stained with Mitotracker Deep Red and transduced with viruses encoding mCherry-FIS1TM and a control shRNA were imaged by time-lapse microscopy. Six examples are shown. The right panel is a magnification of the box in the left panel. At t = 0, force was applied approximately at the center of the red ring. Force was released by AFM tip retraction at time points when the red ring turns green. Blue and yellow arrowheads mark the constricted mitochondria before and after fission respectively. This movie relates to *Figure 2C*. Scale bar, 2 μm.
DOI: https://doi.org/10.7554/eLife.30292.011

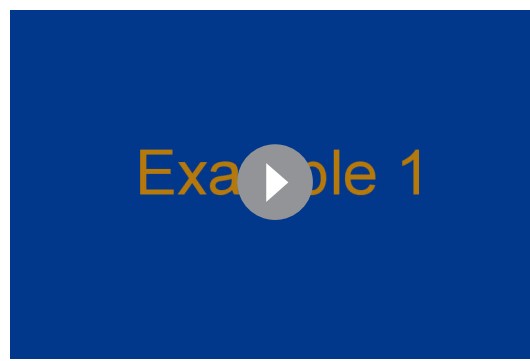

**Video 7.** Mitochondria in DRP1-deficient cells mechanically stressed by contact with an AFM tip. As in *Video 6* except that cells were treated with virus encoding DRP1 shRNA. This movie relates to *Figure 2D*. Scale bar, 2 μm.
DOI: https://doi.org/10.7554/eLife.30292.012

surfaces. The surface of gramophone records consists of grooves 40 μm deep and 80 μm wide approximately, and flat ledges between the grooves (*Read, 1952*) (*Figure 3A*). We hypothesized that, when grown on vinyl records, the spreading of the cell across the ledge over into the groove would cause the peripheral cytoplasmic content to be constricted along the edge. To verify this, we imaged cells expressing an ER marker – GFP-Sec61β – as a proxy for the cytoplasm (*Kanfer et al., 2015*). Indeed, as shown in *Figure 3B* and *Video 9*, 3D-reconstruction of a cell spreading over the edge shows that the cytoplasm is constricted right over the edge.

We anticipated that these constricted sites would be hot-spots for mitochondrial fission. Indeed, the cytoplasmic areas of cells spanning the edge of the groove were usually devoid of mitochondria (*Figure 3C* scrambled siRNA and wild-type).

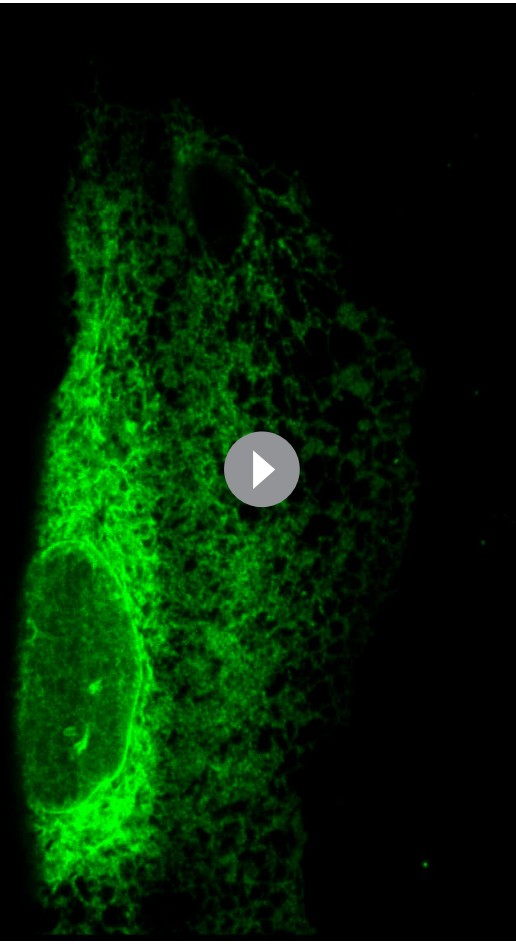

**Video 9.** 90-degree tilting of a 3D-reconstructed KERMIT cell expressing Sec61β-GFP, grown on a vinyl record. This movie relates to *Figure 3B*.
DOI: https://doi.org/10.7554/eLife.30292.015

To assess whether these mitochondria-free regions resulted from increased mitochondrial fission, we used time-lapse microscopy, and focused on the rare cells that still had intact mitochondria spanning the edge. Such cells may have just moved over the groove and their mitochondrial network may not yet have had the time to readjust. Indeed, a few minutes into time-lapse recording, mitochondria vacated the edge area by first undergoing fission (averagely within 93 s), and then moving towards the ledge and the groove areas (*Figure 3E* and *Video 10*). The majority of mitochondria on the ledge area did not undergo fission within an average of 765 s (time in which they remained in focus, or until the end of the microscopy session). Thus, mitochondrial fission happens at places where the cell is constricted from spreading on the patterned surface.

We observed a complete division of the mitochondrial network along the edge of the groove

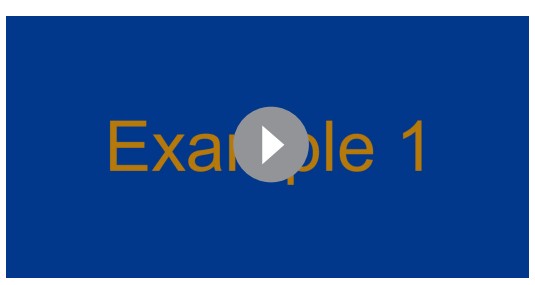

**Video 8.** Mitochondria in DRP1-deficient cells mechanically stressed by contact with an AFM tip. This movie relates to *Figure 2—figure supplement 1*.
DOI: https://doi.org/10.7554/eLife.30292.013

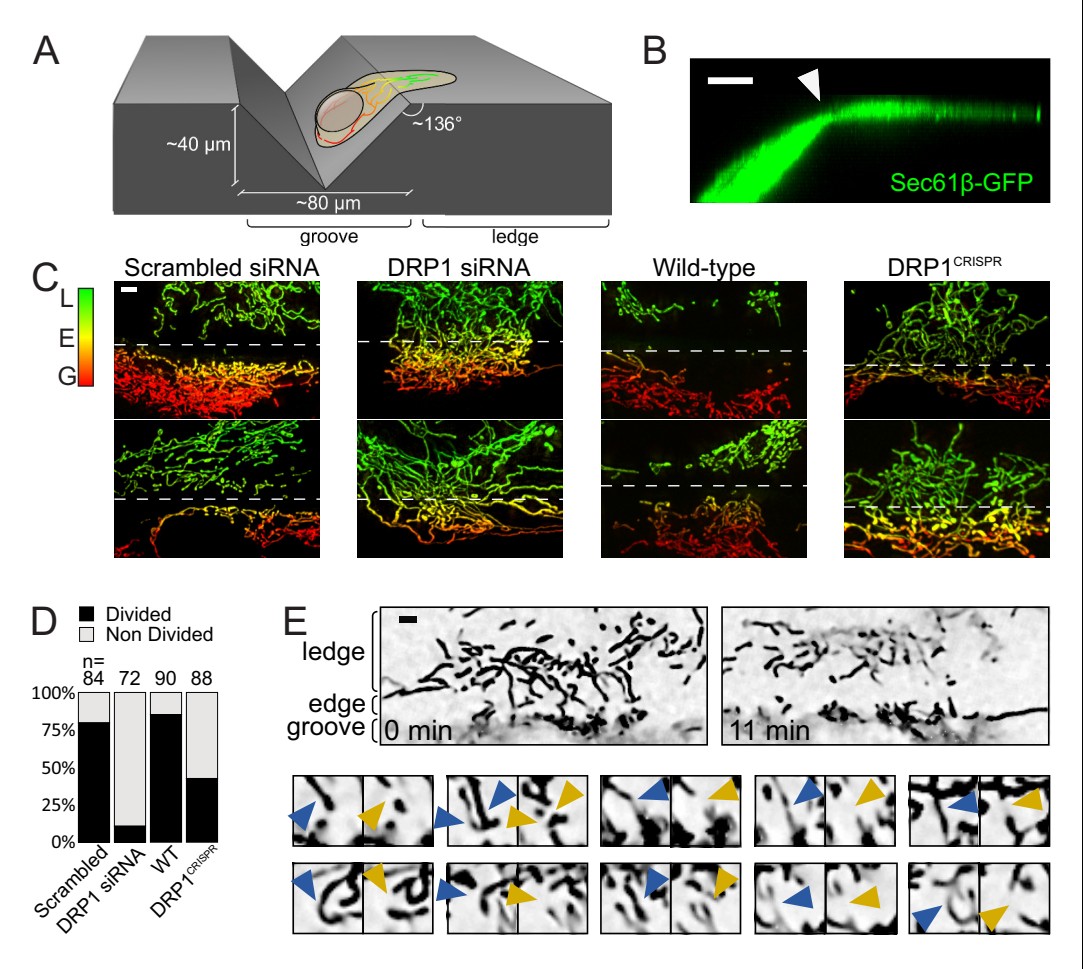

**Figure 3.** Mitochondrial fission upon cell deformation. (**A**) Scheme of the experimental setup. (**B**) 3D projection of a U2OS cell expressing Sec61β-GFP showing thinning of the cytoplasm at the groove's edge (arrowhead). (**C**) Mitochondria of indicated cells grown on vinyl records. Mitochondria are color-coded according to their Z-position (red in the groove, green on the ledge). The dashed lines indicate the approximate position of the edge. (**D**) Quantification of the number of cells showing a divided mitochondrial network, defined as having no mitochondria spanning the edge between the groove and the ledge. Number of cells analyzed are indicated on each bar. (**E**) Time-lapse microscopy of the fission events leading to divided mitochondrial network. Top panel, low magnification of the start- and end-points of the recording. Lower panels, individual fission events captured during the time course. Blue arrowheads, mitochondria before fission; Orange arrowheads, mitochondria after fission. Scale bars, 5 μm.

DOI: https://doi.org/10.7554/eLife.30292.014

in ~80% of wild-type cells, but only in 15% and 43% of the DRP1 siRNA-treated and DRP1[CRISPR] knockout cells, respectively (*Figure 3D*, $p < 10^{-16}$ and $p < 10^{-9}$, from a Fisher's exact test), indicating that the canonical fission machinery is important for this phenomenon. The weaker effect of CRISPR-mediated DRP1 knockout as compared to siRNA-mediated knockdown may be a consequence of possible adaptations to long-term DRP1 depletion.

Thus, mitochondria undergo DRP1-mediated fission in constricted cytoplasmic regions caused by cell adhesion to uneven surfaces.

## Inhibiting ER or actin dynamics does not detectably affect force-induced fission

In unstimulated conditions, ER tubules mark mitochondrial fission sites (*Friedman et al., 2011*). It has thus been proposed that the ER wraps around and constricts mitochondria prior to fission, likely

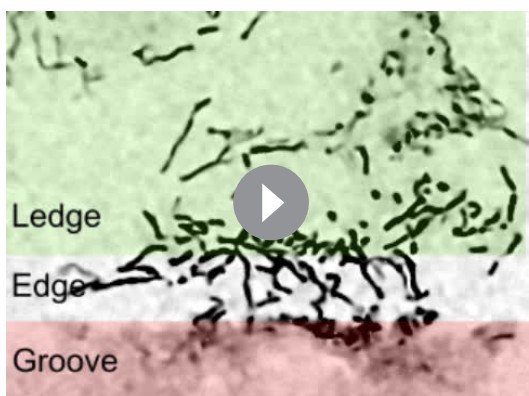

**Video 10.** Time-lapse recording of a KERMIT cell cultured on a vinyl record. The groove and ledge areas are indicated at the beginning and the end of the movie. In the second part of the movie, fission events are indicated with arrowheads (blue, before fission; orange, after fission). This movie relates to *Figure 3E*.
DOI: https://doi.org/10.7554/eLife.30292.016

by locally polymerizing actin via the activities of INverted Formin 2 (INF2) (*Korobova et al., 2013*) and Spire1C (*Manor et al., 2015*). However, when performing AFM stimulation of U2OS cells stably expressing both a mitochondrial (mtBFP) and an ER (GFP-Sec61β) fluorescent marker, we observed that the tip-mediated indentation of cells pushed the ER away from the site of force application and subsequent mitochondrial fission (*Figure 4A*, *Video 11*). This did not appear consistent with an important role of the ER in force-induced fission. To investigate this further, we observed the ER together with mitochondria, in conditions where ER dynamics was perturbed by the overexpression of dominant-negative Atlastin (ATL) mutants or of CLIMP-63. ATLs are large GTPases that mediate ER tubule fusion by forming trans-homooligomers on the cytoplasmic surface of the ER. Overexpression of a GTPase-dead mutant (e.g. ATL1-K80A) or the cytoplasmic domain alone (cyto-ATL2) have a dominant negative effect on endogenous ATLs, preventing ER tubules from fusing with each other (*Goyal and Blackstone, 2013*; *Pawar et al., 2017*). CLIMP-63, on the other hand, stabilizes ER sheets and its overexpression converts most ER tubules to sheets (*Goyal and Blackstone, 2013*).

As expected, the ER in cells overexpressing ATL1-K80A or cyto-ATL2 adopted a hair-like, hypoconnected morphology, increasing the amount of cytoplasmic areas devoid of ER. Conversely, in CLIMP-63-overexpressing cells, the ER was converted from tubules to sheets (*Figure 4B–D*, *Videos 12–14*). The mitochondrial network in these cells showed a normal morphology. In cells found at the edge region of the vinyl grooves, the mitochondria underwent fission along the edge, as in wild-type cells (*Figure 3*). Some of these fission events occurred in areas with only ER sheets (*Figure 4B*, *Video 12*) or without ER at all (*Figure 4C*, *Video 13*). Using the same metrics as in *Figure 3D*, we observed that 89% and 86% of the cells overexpressing ATL1-K80A and CLIMP-63, respectively, displayed a complete separation of their mitochondrial network. These numbers are not significantly different from wildtype cells. Finally, we also infected cells expressing cyto-ATL2 with *Shigella* and observed motile *Shigella*-induced mitochondrial fission in areas devoid of ER signal (*Figure 4D*, *Video 14*). Thus, while the ER might have a function in force-induced fission, this function does not appear strictly necessary.

We also investigated the role of actin in mechanically stimulated mitochondrial fission because actin polymerization is reportedly a crucial downstream step in ER tubule-induced fission, through the ER-anchored isoform of the Inverted Formin 2 (INF2). Knocking-down the ER-associated isoform of INF2 (*Figure 4—figure supplement 1*) did not have a visible effect on collision-induced fission in *Shigella*-infected cells (*Video 15*) or cells grown on gramophone records. Indeed, of the 32 collisions observed in INF2-deficient cells, 25 led to a fission event, a proportion that is in line with that observed in scrambled siRNA-treated cells. Interestingly, actin-based *Shigella* motility was not affected by INF2 depletion, probably because *Shigella* use N-WASP and Arp2/3 to promote actin polymerization (*Gouin et al., 1999*; *Ray et al., 2009*). Similarly, 83% of the cells grown on gramophone records and showing a constricted cytoplasm had separated mitochondrial populations (n = 35), again comparable to what we have observed in control cells. These results suggest that the INF2-mediated pathway does not play essential roles in mechanically induced fission. To address more globally if actin fibers were necessary for force-induced fission, we treated cells with cytochalastin D (cytD), a commonly used drug that inhibits actin polymerization. Because the actin cytoskeleton is necessary for *Shigella* motility and for the proper spreading of cells on vinyl disks, we could only disrupt the actin cytoskeleton in AFM experiments. At effective concentration, the drug severely affected the actin cytoskeleton, as assessed using LifeAct-GFP (*Figure 4E*). In these conditions, the

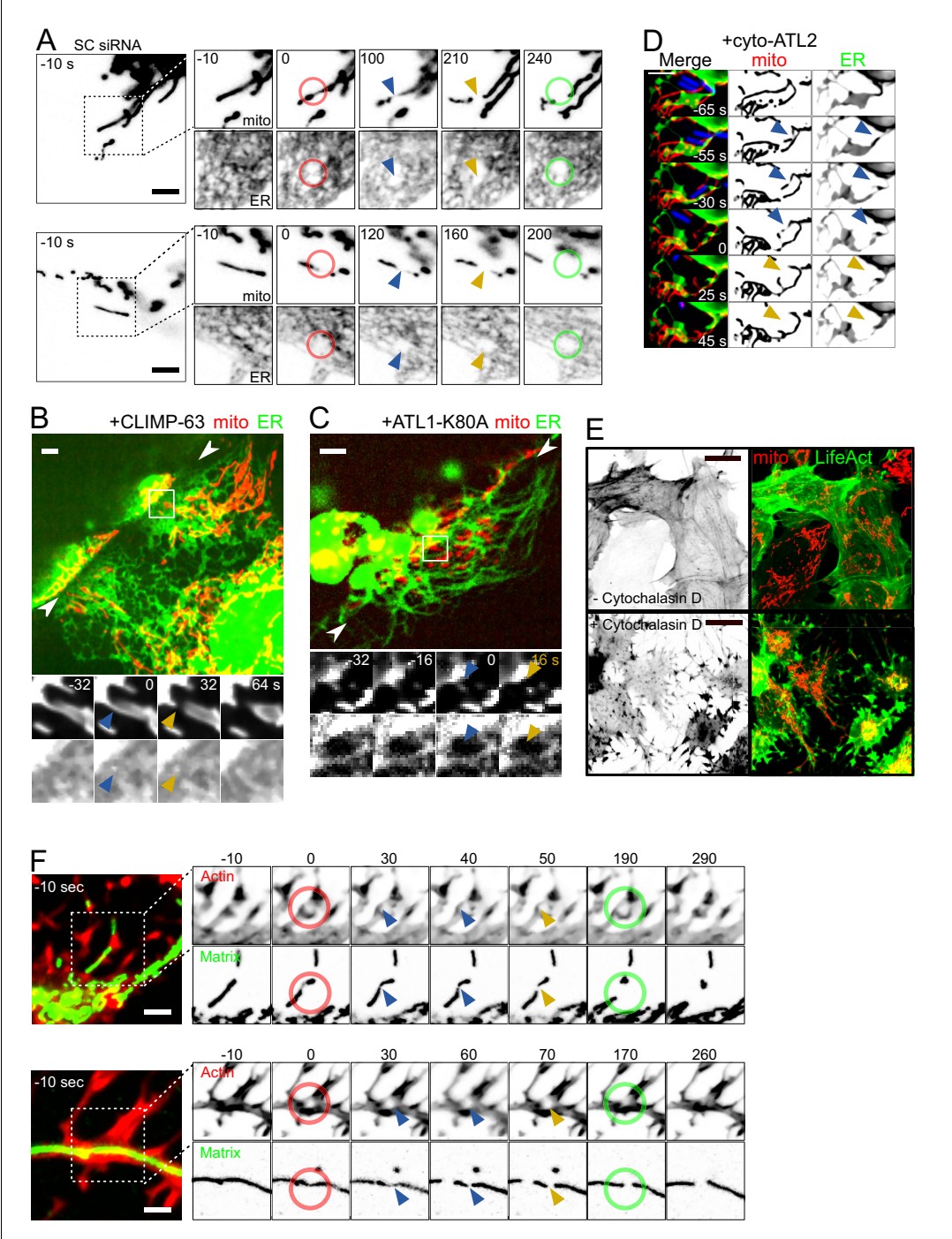

**Figure 4.** Force-induced mitochondrial fission upon ER dynamics perturbations. (**A**) U2OS KERMIT cells stably expressing mtBFP and Sec61β-GFP and treated with scrambled siRNA were imaged by time-lapse microscopy. Two examples are shown. At t = 0 s, the cantilever of the AFM approached the cell in Contact mode, with a force set at 15 nN, at the position of the red ring. Green rings mark the time and area of tip retraction. Blue arrowheads indicate mitochondria that are visibly thinned by the pressure but have not yet undergone fission. Fission events are indicated by an orange arrowhead. (**B**) Cells were seeded on vinyl records and transfected with a CLIMP-63-overexpressing plasmid, effectively converting large fractions of the ER to sheets. Mitochondria underwent fission when spanning over the edge of the groove whether or not the ER had been converted to sheets at the site of fission. The two facing arrowheads indicate the position of the groove's edge. (**C**) As in (**B**) but cells were instead transfected with a construct expressing a dominant-negative form of ATL1-K80A, which inhibits ER interconnection and increases the size of gaps in the ER network. As a result, mitochondria can be observed undergoing fission at sites devoid of ER while spanning the edge of the groove. (**D**) U2OS KERMIT cells were transduced with a lentivirus expressing RFP-Lifeact (blue) and transfected with a cyto-ATL2 expression plasmid. Cells were then infected with mCherry-

*Figure 4 continued on next page*

*Figure 4 continued*

labelled *S. flexneri* (blue). Mitochondria can be observed undergoing fission at sites stimulated by motile bacterium. The blue and yellow arrowheads represent mechanically constricted sites before and after fission, respectively. Mito panels, mtBFP. ER panel, Sec61β-GFP. Scale bars 5 μm. (**E**) U2OS cells transduced with GFP-Lifeact and matrix-targeted RFP were treated with 5 μg/μl of Cytochalasin D for 90 min. (**F**) U2OS cells transduced with GFP-Lifeact and matrix-targeted RFP were treated for 90 min with 1 μg/μl (upper panel) or 5 μg/μl Cytochalasin D (lower panel), respectively. At t = 0 s, the cantilever of the AFM approached the cell in Contact mode, with a force set at 15 nN, at the position of the red ring. Green rings mark the time and area of tip retraction. Blue arrowheads indicate mitochondria that are visibly thinned by the pressure but have not yet undergone fission. Fission events are indicated by an orange arrowhead.

DOI: https://doi.org/10.7554/eLife.30292.017

The following figure supplement is available for figure 4:

**Figure supplement 1.** Control for INF2 knockdown efficiency.

DOI: https://doi.org/10.7554/eLife.30292.018

proportion of mechanically-induced mitochondrial fission was similar to untreated cells (16 fission events from 17 touchdowns, *Figure 4F*, *Video 16*).

Thus, actin polymerization does not appear to be a necessary step in force-induced fission.

## Mechanism of force sensing and DRP1 recruitment

Our observations indicated that mechanical force caused the recruitment and activation of the fission effector DRP1, and raised the question of how such mechanical stimulus was sensed at the molecular level. DRP1 is a cytosolic protein, and is recruited to mitochondria by integral mitochondrial membrane adaptor molecules such as mitochondrial fission factor (MFF) (*Gandre-Babbe and van der Bliek, 2008*) and mitochondrial dynamic protein of 49/51 kDa (Mid49/51) (*Palmer et al., 2011*; *Zhao et al., 2011*). These adaptors are therefore thought to recruit DRP1 to presumptive fission sites (*Friedman et al., 2011*). MFF, for instance, accumulates on dotted structures on mitochondria independently of the presence of DRP1 (*Otera et al., 2010*). It was therefore tempting to hypothesize that adaptor proteins acted directly as mechano-sensors on the mitochondrial surface. To test this idea, we examined the localization of MFF on the mitochondrial surface by immunofluorescence. We performed these experiments in the absence of DRP1 to exclude the possibility that DRP1 may influence MFF localization. As previously reported (*Otera et al., 2010*), MFF localized to foci irrespective of the presence of DRP1 (*Figure 5—figure supplement 1A*). Additionally, we observed that MFF had a tendency to accumulate at constrictions that happen sporadically on non-

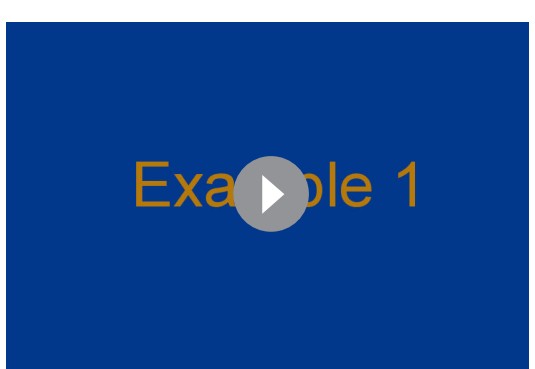

**Video 11.** The AFM tip displaces the ER during force-induced mitochondral fission. U2OS KERMIT cells stably expressing mtBFP and Sec61β-GFP and treated with scrambled siRNA were imaged by time-lapse microscopy. Three examples are shown. At t = 0 s, the cantilever of the AFM approached the cell in Contact mode, with a force set at 15 nN, at the position of the red ring. Green rings mark the time and area of tip retraction. Blue arrowheads indicate mitochondria that are visibly thinned by the pressure but have not yet undergone fission. Fission events are indicated by an yellow arrowhead. This movie relates to *Figure 4A*.

DOI: https://doi.org/10.7554/eLife.30292.019

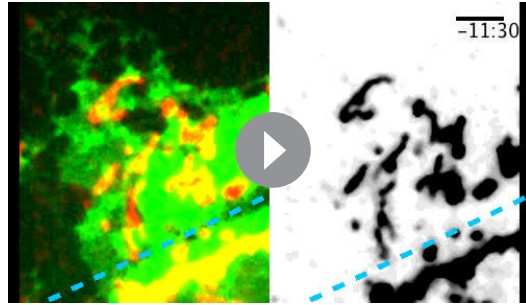

**Video 12.** Force-induced mitochondrial fission in cells overexpressing CLIMP-63. KERMIT cells were seeded on vinyl records and transfected with a CLIMP-63-overexpressing plasmid. Fission events along the groove edge are indicated with arrowheads (blue, before fission; orange, after fission). Blue dashed line at the beginning of each video indicates the position of the edge. Time stamp is relative to the first fission event in each movie. This video relates to *Figure 4B*.

DOI: https://doi.org/10.7554/eLife.30292.020

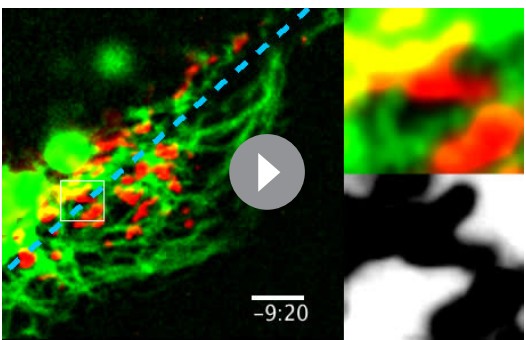

**Video 13.** Force-induced mitochondral fission in cells overexpressing ATL1-K80A. KERMIT cells were seeded on vinyl records and transfected with an ATL1-K80A-overexpressing plasmid. Fission events along the groove edge are indicated with arrowheads (blue, before fission; orange, after fission). Blue dashed line at the beginning of each video indicates the position of the edge. Time stamp is relative to the first fission event in each movie. This video relates to *Figure 4C*.
DOI: https://doi.org/10.7554/eLife.30292.021

perturbed mitochondrial tubule (*Figure 5A*, arrowheads in right panels). This propensity was already observed previously in live cells using GFP-tagged MFF (*Friedman et al., 2011*). To assess if MFF would localize to force-induced constrictions, we again cultured DRP1-depleted cells on gramophone records, and observed MFF on constricted mitochondria at the groove edge. In the majority of the cases where mitochondria were visibly constricted, we observed MFF foci right on or at the edge of constrictions (*Figure 5B*, *Figure 5—figure supplement 1B*), suggesting that MFF senses mechanical constrictions on mitochondria upstream of DRP1. In line with these observations, we observed signifi-

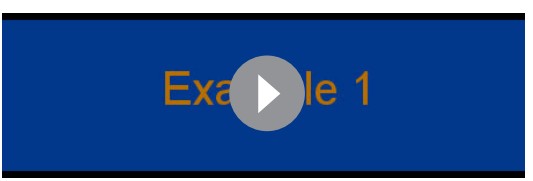

**Video 15.** INF2-independent mitochondrial fission induced by *Shigella*. U2OS cells transduced with lentiviruses expressing mtBFP and GFP-Lifeact were reverse-transfected with scrambled (SCR) siRNA or siRNA specifically targeting INF2-CAAX isoform. 72 hr later, they were infected with GFP-labelled *S. flexneri*. Red, mitochondrial matrix. Green, *Shigella* and actin. Blue and orange arrowheads indicate mitochondria before and after *Shigella* -induced fission, respectively. Scale bar, 2 μm.
DOI: https://doi.org/10.7554/eLife.30292.023

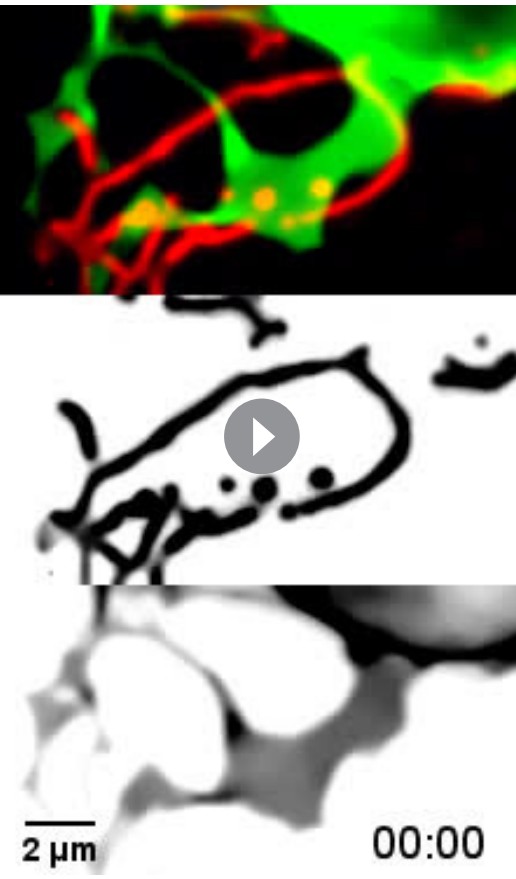

**Video 14.** *Shigella*-induced mitochondral fission in cells overexpressing cyto-ATL2. U2OS KERMIT cells were transduced with a lentivirus expressing RFP-Lifeact (blue) and transfected with a cyto-ATL2 expression plasmid. Cells were then infected with mCherry-labelled *S. flexneri* (blue). Mitochondria can be observed undergoing fission at sites stimulated by motile bacterium. Blue and yellow arrowheads represent mechanically constricted sites before and after fission, respectively. Mito panels, mtBFP. ER panel, Sec61β-GFP. Scale bars, 2 μm. This video relates to *Figure 4D*.
DOI: https://doi.org/10.7554/eLife.30292.022

cantly fewer fission events in MFF-deficient cells (*Figure 1—figure supplement 1D*) as compared to wild-type cells using the *Shigella* as well as the vinyl disk models (5.0% vs 56.5% and 38% vs 80%, respectively; $p < 10^{-4}$ and $p < 10^{-9}$ from Fisher's exact tests, respectively).

To observe this phenomenon in live cells, we infected cells with lentiviruses expressing GFP-tagged MFF to assess its localization upon either *Shigella*- or vinyl disk-triggered mechanical stimulation. In these experiments, we again downregulated DRP1 expression using siRNAs. Successful constriction of mitochondria was monitored by the reduction of the mtBFP signal.

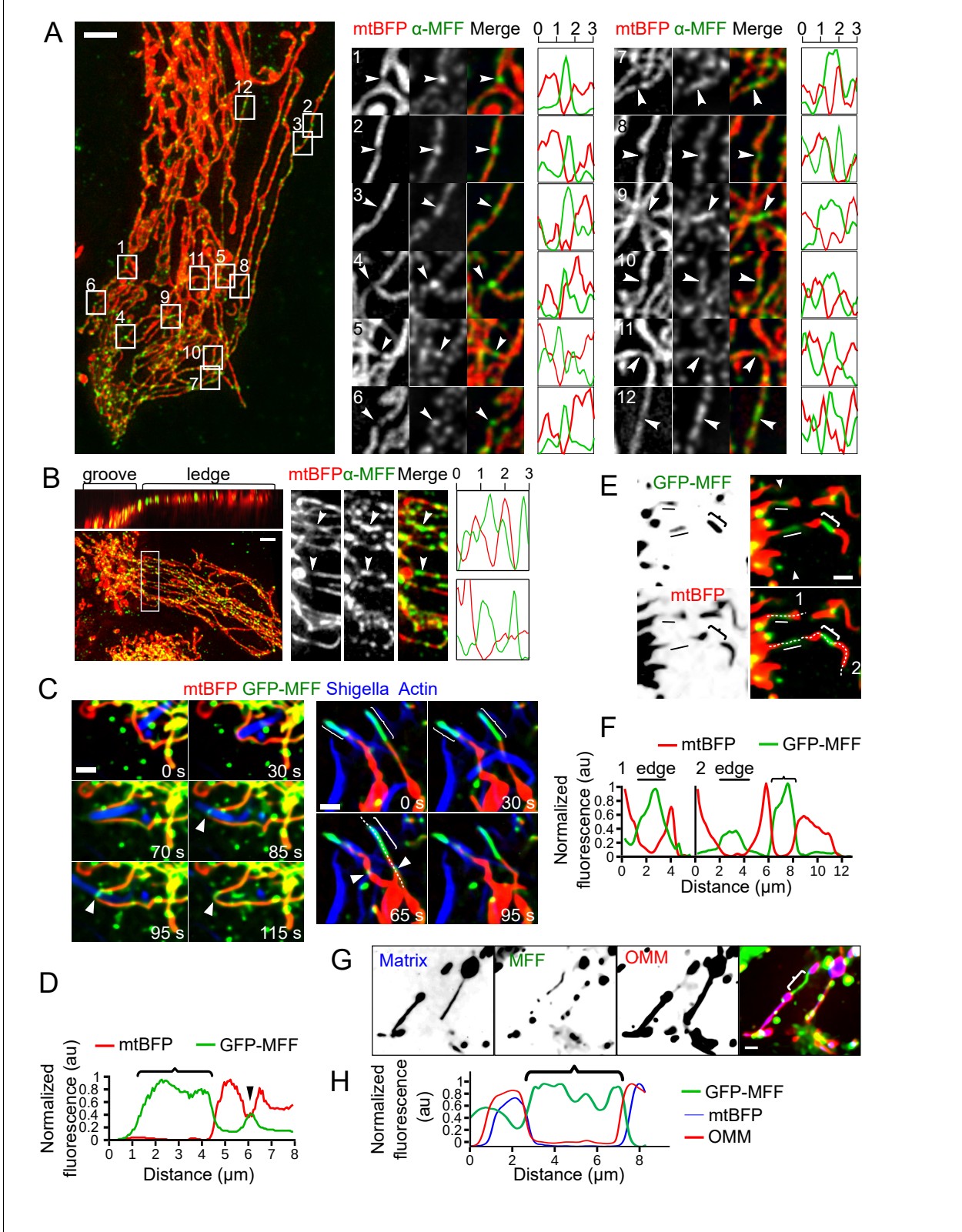

**Figure 5.** MFF recruitment to mechanically strained sites. (**A**) Immunofluorescence of Kermit cells transduced with shDRP1, using an anti-MFF antibody (green). Mitochondrial matrix (mtBFP) is shown in red. Insets on the right correspond to the framed areas on the left. Arrowheads point at naturally occuring constrictions on the mitochondria. Plots are linescans of the mitochondria (red) and MFF (green) signals around the constriction. X-axis is in μm. Y-axis is normalized fluorescence in arbitrary units. (**B**) as in A, but the cells were grown on gramophone records. Top left panel is a 3D projection

*Figure 5 continued on next page*

*Figure 5 continued*

of the same cell rotated by 90° around the x-axis. The white box on the left corresponds to the magnified area on the right. Arrowheads indicate mitochondrial constrictions at the groove's edge. (C) U2OS cells were transduced with lentiviral vectors expressing mtBFP, RFP-Lifeact, GFP-MFF as well as shRNA targeting DRP1. Puromycin-resistant cells (shRNA-positive) were then infected with RFP-labeled *S. flexneri*, and imaged by time-lapse microscopy. MFF is recruited to sites of encounter with *S. flexneri* (white arrowheads). Right panels also show two examples of MFF enrichment at sites of mitochondria thinning (curly brackets), as indicated by reduction of matrix mtBFP signal, independent of *Shigella* encounter. (D) Line scan of mtBFP and MFF signal of the white dotted line in (C). Arrowhead and curly bracket correspond to same zones in (C). Normalized background-subtracted pixel values are plotted as arbitrary units. (E) Z-projected image of a transduced U2OS cell spanning the edge of vinyl groove expressing mtBFP, GFP-MFF as well as shRNA targeting DRP1. The groove's edge is indicated by two facing arrowheads on top right panel. Two stabilized individual mitochondrial tubules span over the edge (white line), and show loss of matrix BFP signal and increased MFF signal. Another example of GFP-MFF enrichment to a constricted mitochondrial tubule outside of the edge area is indicated by a curly bracket. (F) Dotted white lines 1 and 2 in (E) are selected for line plots as in (D). (G) U2OS cells were transduced with lentiviral vectors expressing mtBFP, mCherry-Fis1TM (OMM), as well as shRNA targeting DRP1. Puromycin-resistant cells (shRNA-positive) were then transfected with GFP-MFF, and imaged by time-lapse microscopy. MFF spontaneously stabilizes thin mitochondrial section (curly brackets) that are devoid of matrix staining but retain continuous OMM signal. (H) Line scan of mtBFP, GFP-MFF and OMM signal of the curly bracket in (G). Scale bars, A-B, 5 μm, C-H 2 μm.

DOI: https://doi.org/10.7554/eLife.30292.025

The following figure supplement is available for figure 5:

**Figure supplement 1.** MFF immunostaining in WT and DRP1-deficient cells.
DOI: https://doi.org/10.7554/eLife.30292.026

GFP-MFF was recruited to constricted mitochondria in both model systems (*Figure 5C–D*, arrowheads, *Video 17*; *Figure 5E–F*).

Thus, MFF shows preferential accumulation on thinned mitochondria and could therefore, serve as a mechano-sensor for the recruitment of the fission machinery.

In the course of the above experiments, we frequently observed mitochondrial regions where MFF accumulated preferentially (*Figure 5C–F*, curly brackets), independent of *Shigella-* or pattern-mediated mechanical stimulation, including in non-stimulated resting cells. These regions showed a reduction of the matrix-targeted mtBFP signal, indicating that they had a reduced mitochondrial diameter. The OMM-directed fluorescence marker (mCh-Fis1TM) showed that these mitochondria retained connectivity. *Figure 5G–H* and *Video 18* show striking examples of this behavior, where thin, GFP-MFF-positive, matrix-negative tubules connected to thick sections of mitochondria. The two sections remained connected for several minutes, but remained stably topologically separated in two distinct mitochondrial domains. Importantly, these structures were only observed when GFP-MFF was overexpressed to relatively

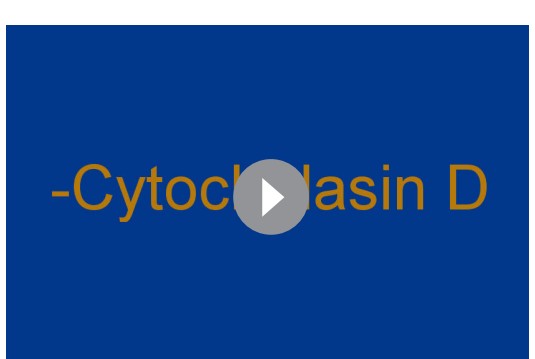

**Video 16.** Actin-independent force-induced mitochondral fission. U2OS cells transduced with GFP-Lifeact and matrix-targeted RFP were treated for 90 min with Cytochalasin D (Examples 1, 3, 5 at 1 μg/μl; Examples 2, 4 at 5 μg/μl). At t = 0 s, the cantilever of the AFM approached the cell in Contact mode, with a force set at 15 nN, at the position of the red ring. Green rings mark the time and area of tip retraction. Blue arrowheads indicate mitochondria that are visibly thinned by the pressure but have not yet undergone fission. Fission events are indicated by an orange arrowhead. This video relates to *Figure 4F*.
DOI: https://doi.org/10.7554/eLife.30292.024

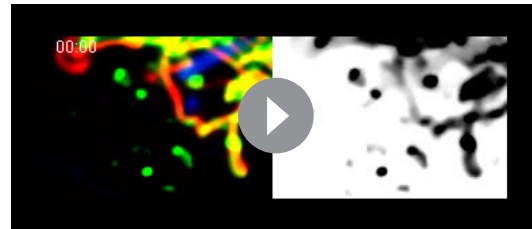

**Video 17.** MFF recruitment to mitochondria constricted by the encounter with *Shigella*. U2OS cells transduced with lentiviruses expressing RFP-Lifeact, mtBFP, GFP-MFF and DRP1-specific shRNA were infected with RFP-labelled *S. flexneri*. Red, mitochondria. Blue, *Shigella* and actin. Green, GFP-MFF. Arrowheads indicate MFF recruitment to constricted mitochondria. This movie relates to *Figure 5C*.
DOI: https://doi.org/10.7554/eLife.30292.027

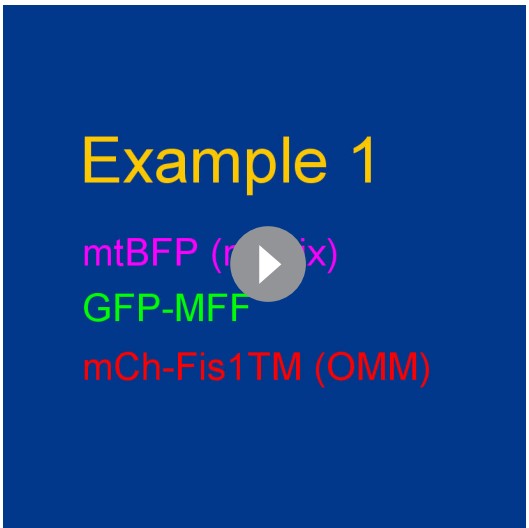

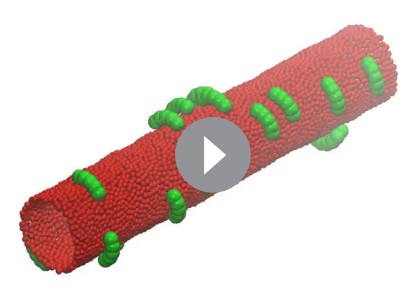

**Video 18.** High level MFF overexpression stabilizes thin, matrix-free mitochondria. U2OS cells were transduced with lentiviral vectors expressing mtBFP, mCherry-Fis1TM (OMM), as well as shRNA targeting DRP1. Puromycin-resistant cells (shRNA-positive) were then transfected with GFP-MFF, and imaged by time-lapse microscopy. MFF spontaneously stabilizes thin mitochondrial section (white arrowheads) that are devoid of matrix staining but retain continuous OMM signal. This movie relates to *Figure 5G*.
DOI: https://doi.org/10.7554/eLife.30292.028

**Video 19.** Monte Carlo simulation of low density of protein on a membrane tube. Proteins and membranes were modelled as in *Figure 6*. 20 proteins with an $R_{pr}$ of 3.5 $\sigma$ were allowed to reach equilibrium on a membrane tube with a radius of 10 $\sigma$ and a length of 100 $\sigma$.
DOI: https://doi.org/10.7554/eLife.30292.031

high levels, indicating that they resulted directly or indirectly from this overexpression. Thus, it appeared that MFF not only acts as a sensor but could also potentially act as an inducer of mitochondrial constriction.

Because it seemed paradoxical that MFF behaved as both a sensor and inducer of mitochondrial constrictions, we wondered whether these two properties may be coupled. To test this hypothesis, we turned to computer-assisted Monte Carlo (MC) simulations and modelled a generic protein with an affinity for constricted mitochondria. To simulate the desired affinity, we designed a protein with a curved membrane-binding surface (*Figure 6A*), and placed several copies of it on a membrane tube, the radius of which was larger than that of the protein's binding surface. After finding the optimal ratio of the protein and tube diameters that led to protein assembly (*Figure 6—figure supplement 1*), we performed a first simulation, where we used an arbitrarily low protein density. Under such conditions, the protein diffused freely on the membrane tube and remained homogenously distributed (*Figure 6B*, *Video 19*). We then repeated the simulation, but this time, we pre-imposed a constriction in the membrane tube, to mimic mechanical stimulation, and observed that the proteins accumulated at the constriction site (*Figure 6C*, *Video 20*), resembling MFF recruitment to mechanically strained sites (*Figure 5C–F*, *Video 17*). Next, we repeated the simulation with a high protein density. Interestingly, under these conditions the proteins spontaneously constricted the membrane tube even without pre-imposed constriction (*Figure 6D*, *Video 21*). The results of the simulations strongly resembled the MFF-stabilized, matrix-free thin mitochondria sections we observed upon high levels of MFF overexpression (*Figure 5C–H*, *Video 18*).

## Discussion

Our study shows that mitochondria do not only respond to biochemical, but also to mechanical cues; mechanical force, via the deformation of mitochondrial membranes, leads to mitochondrial fission, coupling a mechanical trigger to a biochemical response.

How is this response orchestrated at the molecular level? We report a yet-unknown property of MFF – a preference for mitochondrial tubules of smaller diameter. Thus, a straightforward model for

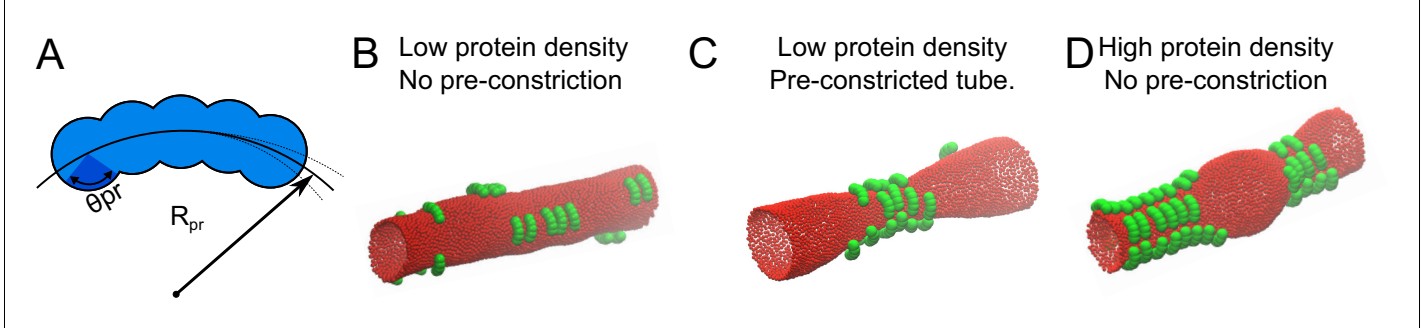

**Figure 6.** Monte Carlo simulation of protein-membrane interactions for different conditions. (A) Proteins were modelled as a linear chain built out of five spheres positioned on a circular arc. Each sphere has a radius of σ. The center-to-center distance between the spheres within a protein was adjusted to 2/3 of σ. $R_{pr}$, is the radius of the arc, for which we found the optimal value to be 3.5σ (see supplement), and $\theta_{pr}$ is the protein-membrane contact angle, which we set to π/4, in order for the proteins to attract the membrane only by the inner part of their structure. (B) 20 proteins as in A with an optimized $R_{pr}$ were allowed to reach equilibrium on a membrane tube with a radius of 10σ and a length of 100σ. (C) 20 proteins as in A were simulated on a membrane tube with a pre-constriction (radius at the center of the constriction = 3σ) and allowed to equilibrate. (D) 50 proteins as in A were allowed to reach equilibrium on a membrane tube as in B (without pre-constriction).

DOI: https://doi.org/10.7554/eLife.30292.029

The following figure supplement is available for figure 6:

**Figure supplement 1.** Optimization of the binding curvature of the protein during Monte Carlo simulation.

DOI: https://doi.org/10.7554/eLife.30292.030

force-induced fission is that physical constriction might reduce mitochondrial diameter, causing MFF accumulation, subsequent DRP1 recruitment and mitochondrial fission.

We also observe that MFF acts not only as a sensor but also as an inducer of constrictions. As shown by our Monte-Carlo simulations, a protein with increased affinity for mitochondria of reduced diameter will rapidly accumulate at constricted sites. At high concentrations however, it will induce and stabilize constrictions, two behaviors that mimic what we observe for GFP-MFF in live cells. Therefore, the propensity of a protein to localize to constricted sites is intrinsically coupled to its ability to stabilize constrictions. The difference between the inducing and sensing behaviors seems

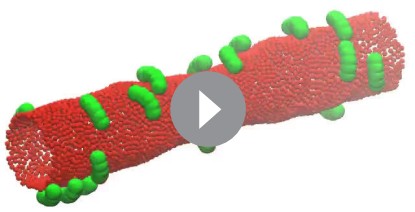

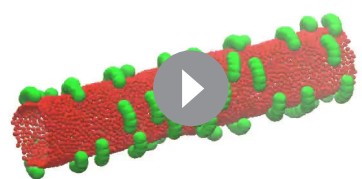

**Video 20.** Monte Carlo simulation of low density of protein on a membrane tube with a pre-constriction site. Proteins and membranes were modelled as in *Figure 6* and simulated as in *Video 19*, except that a pre-constricted site was present on the membrane tube throughout the simulation.

DOI: https://doi.org/10.7554/eLife.30292.032

**Video 21.** Monte Carlo simulation of high density of protein on a membrane tube. Proteins and membranes were modelled as in *Figure 6* and simulated as in *Video 19*, except that 50 instead of 20 proteins were present on the membrane tube.

DOI: https://doi.org/10.7554/eLife.30292.033

to lie in the quantity of the protein. Indeed, MFF only stabilized constrictions when highly overexpressed. In fact, similar observations have been made for membrane curvature-sensing proteins (*Baumgart et al., 2011*; *Simunovic et al., 2015*; *Sorre et al., 2012*). Of note, the protein simulated here is not meant to reflect the shape of MFF. Our simulations however highlight that sensing and inducing constrictions are two sides of the same coin, and that the difference between sensing and inducing lies in the expression level of the factor. Actually, it is unknown if MFF monomers or oligomers have a curved membrane-binding interface, like our model protein in the simulations. MFF affinity for constricted mitochondria may stem from other determinants such as affinity for specific lipid species, or properties of its transmembrane domain. Several other DRP1 adapters exist in the cell, hinting at a diversity of determinants for DRP1 recruitment. In this context, MFF might not be the only force sensor on the mitochondrial surface. Similarly, force is unlikely to be the only trigger for mitochondrial fission.

In unperturbed adherent cells, mitochondria fission happens almost exclusively at sites in contact with the ER (*Friedman et al., 2011*). Force-induced mitochondrial fission may provide a mechanistic explanation for this phenomenon. Indeed, adherent cells have a very flat periphery (*Xu et al., 2012*) where a single layer of ER tubules is present, making them especially suited for microscopy studies. For this reason, ER-mediated mitochondrial division has mostly been studied in peripheral areas of adherent cells. Mitochondrial tubules are generally larger in diameter (~500–700 nm) (*Youle and Karbowski, 2005*) than the thickness of the cytoplasm in these peripheral regions. Indeed, super-resolution imaging studies show that peripheral mitochondria are flattened by the proximity of the 'ventral' and 'dorsal' actin cortices (*Huang et al., 2008*; *Huang et al., 2016*), adopt an elliptic – rather than tubular – cross-section, and slightly bulge from the cell surface (*Wojcik et al., 2015*). In live-cell high-speed AFM experiments, mitochondria are seen to deform the plasma membrane and underlying actin cytoskeleton by ~ 50 nm (*Yoshida et al., 2015*). It is therefore clear that peripheral mitochondria, sandwiched between the 'dorsal' and 'ventral' actin cortices, are mechanically constrained in these areas. Why does the pressure then not fragment the peripheral mitochondria? Probably because the pressure from the actin cortices applies uniformly over the whole length of the mitochondria and does not create local constrictions where MFF and DRP1 could enrich. In this context, it is tempting to speculate that crossing-over ER tubules, rigidified by an INF2- and Spire1C-mediated actin shell, might impose localized mechanical force to underlying mitochondria, triggering fission at ER-mitochondria encounter sites. Being the most extended intracellular membrane network, the ER is also the most likely to physically clash with the mitochondrial network, explaining the frequent observation of fission at ER contact sites. In support of this idea, we observe that mechanically-stimulated fission does not appear to be dependent on ER tubules. An appealing hypothesis is that the ER constitutes an intrinsic source of mechanical force. It therefore appears dispensable in our experimental conditions because an extrinsic source of mechanical force is present. This, in turn, suggests that for ER-mediated fission events, the ER may be imposing force rather than providing biochemical signals. Because actin is an important force-generator in the cell, our model might also explain why it is necessary for mitochondrial fission (*Korobova et al., 2013*; *Manor et al., 2015*), unless cells are mechanically stimulated with an AFM (*Figure 4E–F*).

The role of mitochondrial fusion and fission has remained mysterious. Both processes have been proposed to serve in the repair of damaged mitochondria by fusing with healthy ones, in the segregation of terminally damaged mitochondria for disposal by mitophagy, and in the correct apportionment of mitochondria to daughter cells upon mitosis (*Katajisto et al., 2015*; *Twig et al., 2008*). Here, we show that mitochondria undergo DRP1- and MFF-mediated fission in response to mechanical stimulation.

Cells constantly undergo events of organelle transport, cytoskeleton growth and shrinkage, and general cytoplasm remodeling. Organelles as extended as the ER and mitochondria have to cope with an ever-changing environment and avoid clashes and entanglement with each other. By resolving any resulting mechanical strain, fission might enable mitochondria to escape shearing and ripping, (which could lead to Cytochrome-C release and apoptotic cell death), and avoid entanglements. In that capacity, DRP1 activity may be paralleled to that of topoisomerase-II in disentangling the genome. Using serial block-face scanning electron microscopy, we observe concatenations between mitochondria and the ER in mammalian cells (our unpublished observations). Given the dynamic nature of these organelles, entanglements are likely to induce mechanical strain on mitochondria, therefore leading to their resolution by fission. Unlike mitochondrial fission, ER fission

is not frequently observed. Thus, we presume that mitochondrial fission is the prime mechanism to resolve these concatenations. Mechano-induced mitochondrial remodeling could play particularly important roles during cell migration through narrow passages, such as leukocyte extravasation and immune surveillance (*Muller, 2011*; *Nourshargh and Alon, 2014*) as well as cancer cell metastasis (*Friedl and Wolf, 2003*).

In conclusion, our study shows that mechanical force can be coupled to a biochemical process crucial for intracellular membrane shaping and distribution, providing a new perspective in mitochondrial dynamics and organelle interactions.

# Materials and methods

## Key resource table

| Reagent type (species) or resource | Designation | Source or reference | Identifiers | Additional information |
|---|---|---|---|---|
| strain, strain background (*Shigella flexneri,serotype 5a strain M90T*) | GFP-tagged shigella | other | | Jost Enninga (Paris) |
| strain, strain background (*Shigella flexneri,serotype 5a strain M90T*) | RFP-tagged shigella | other | | Jost Enninga (Paris) |
| strain, strain background (*Shigella flexneri,serotype 5a strain M90T*) | mCherry-tagged shigella | PMID:24039575 | | strain only used in S. Mostowy lab (London) |
| cell line (*Homo sapiens*) | U2OS | other | | Matthias Peter (Zurich) |
| cell line (*H. sapiens*) | U2OS-KERMIT | PMID:26259702 | | |
| cell line (*Cercopithecus aethiops*) | COS7 | UCSF cell culture facility | | |
| cell line (*H. sapiens*) | KERMIT-DRP1$^{CRISPR}$ | this paper | | CRISPR-mediated DRP1 knockout cell line, generated using pX330-DRP1e × 2 and pX330-DRP1e × 6 (see entry for these plasmids) |
| antibody | Anti-DRP1 (mouse monoclonal) | Abcam | Abcam:ab56788 | (1:2000) |
| antibody | Anti-MFF (rabbit polyclonal) | SIGMA-ALDRICH | SIGMA:HPA010968 | (1:50) |
| antibody | Anti-MFF (rabbit polyclonal) | Protein Tech Group, Inc. | Proteintech: 17090–1-AP | (1:2000) |
| antibody | Anti-α-Tubulin | SIGMA-ALDRICH | SIGMA:T5168 | (1:2000) |
| recombinant DNA reagent | GFP-DRP1 (plasmid) | PMID:26101352 | | |
| recombinant DNA reagent | Mid49-Cherry (plasmid) | PMID:26101352 | | |
| recombinant DNA reagent | mCherry-DRP1 (plasmid) | PMID:21885730 | | |
| recombinant DNA reagent | mCherry-Lifeact (plasmid) | PMID:22980331 | | |
| recombinant DNA reagent | ATL1-K80A; CLIMP-63 (plasmid) | other | | Robin Klemm (Zurich) |
| recombinant DNA reagent | Cyto-ATL2 (plasmid) | PMID:28826471 | | |
| recombinant DNA reagent | pLVX-puro-GFP-Lifeact; pLVX-puro-RFP-Lifeact; pPax2; pMD2.G (plasmid) | other | | Michael Way (London) |
| recombinant DNA reagent | pLVX-GFP-Lifeact; pLVX-RFP-Lifeact (plasmid) | this paper | | puromycin-resistant cassette deleted. |
| recombinant DNA reagent | pLVX-mtBFP (plasmid) | this paper | | Progentiors: PCR, mtBFP from pcDNA3.1-mtBFP (PMID:26259702); Vector pLVX |
| recombinant DNA reagent | pLVX-Puro-MFF (plasmid) | this paper | | Progenitors: PCR, GFP-MFF (Addgene:49153); Vector pLVX-puro |

*Continued on next page*

*Continued*

| Reagent type (species) or resource | Designation | Source or reference | Identifiers | Additional information |
|---|---|---|---|---|
| recombinant DNA reagent | pLVX-mCherry-Fis1TM (plasmid) | this paper | | Progenitors: PCR, Fis1 transmembrane domain from pBK416 (PMID:28864540); Vector pLVX |
| recombinant DNA reagent | shCtrl (plasmid) | SIGMA-ALDRICH | SIGMA:SHC201 | |
| recombinant DNA reagent | shDRP1 (plasmid) | SIGMA-ALDRICH | SIGMA: TRCN0000318425 | |
| recombinant DNA reagent | pX330-DRP1e × 2 (plasmid) | this paper | | pX330-DRP1e × 2 was generated by cloning the following annealed oligodeoxynucleotides into the pX330 vector (5'-caccGTGACAATTCCAGTACC TCT-3', 5'-aaacAGAGGTACTGGAATT GTCAC-3';) |
| recombinant DNA reagent | pX330-DRP1e × 6 (plasmid) | this paper | | pX330-DRP1e × 6 was generated by cloning the following annealed oligodeoxynucleotides into the pX330 vector (5'- caccGAGACCTCTCATTC TGCAAC-3', 5'- aaacGTTGCAGAATGAG AGGTCTC-3') |
| sequence-based reagent | siDRP1 #1 | PMID:15286177 | | siRNA 5'-UCCGUGAUGAGUAUG CUUUdTdT-3' |
| sequence-based reagent | siDRP1 #2 | PMID:21186368 | | siRNA 5'-CTGGAGAGGAATGCTGAAA-3' |
| sequence-based reagent | siMFF | this paper | | siRNA 5'-CUGAGCAGUUCUGCA GUAACAdTdT-3' |
| sequence-based reagent | siINF2-CAAX | PMID:23349293 | | siRNA 5'-ACAAAGAAACTGTGTGTGA-3' |
| software, algorithm | ndsafir | PMID:19900849 | | |

## Plasmids and cells

GFP/mcherry-DRP1 and Mid49-Cherry have been previously described (*Elgass et al., 2015*; *Friedman et al., 2011*). The plasmid expressing mCherry-Lifeact was previously described in Humphries et al. (*Humphries et al., 2012*). ATL1-K80A and CLIMP-63 overexpression plasmids were kind gifts from Robin Klemm (Zurich). Cyto-ATL2 expression plasmid was a kind gift from Sumit Pawar and Ulrike Kutay (ETHZ). Lentiviral transfer plasmids pLVX-puro-GFP-Lifeact and pLVX-puro-RFP-Lifeact were generous gifts from Michael Way (London). Puromycin-sensitive versions were produced by deleting the puromycin cassette from these vectors to yield pLVX-GFP/RFP-Lifeact. pLVX-mtBFP was constructed by swapping the GFP-Lifeact fragment in pLVX-GFP-Lifeact with the mtBFP fragment in pcDNA3.1-mtBFP (*Kanfer et al., 2015*). pLVX-Puro-MFF was generated by cloning the GFP-MFF fragment from pGFP-MFF (Gia Voeltz; Addgene plasmid # 49153)) into the pLVX-puro vector. pLVX-mCherry-Fis1TM was generated by cloning the transmembrane domain of yeast Fis1 into the pLVX vector. Lentiviral transfer plasmids for control and DRP1-targeting shRNA were purchased from SIGMA (Saint-Louis, MO, SHC201 and TRCN0000318425, respectively). pX330-DRP1ex2 and pX330-DRP1ex6 plasmids were generated by cloning the following annealed oligo-deoxynucleotides into the pX330 vector (for DRP1ex2, 5'-caccGTGACAATTCCAGTACCTCT-3', 5'-aaacAGAGGTACTGGAATTGTCAC-3'; for DRP1ex6, 5'- caccGAGACCTCTCATTCTGCAAC-3', 5'-aaacGTTGCAGAATGAGAGGTCTC-3'), as described (*Cong et al., 2013*). U20S cells were obtained from Matthias Peter's lab, (ETHZ), COS7 cells from the UCSF cell culture facility, and U2OS-KERMIT cells from our collection (*Kanfer et al., 2015*). Cells expressing the indicated fluorescent proteins were maintained in DMEM supplemented with GlutaMAX-I (GIBCO), 0.3 mg/ml L-glutamine, 10% FCS, 100 U/ml penicillin and 100 ug/ml streptomycin in a humidified incubator at 37°C, 5% $CO_2$.

## Antibodies

Anti-DRP1 (ab56788, Abcam, Cambridge, GB) and Anti-α-Tubulin (T5168, Sigma) antibodies were used at a dilution of 1/2000 for immunoblotting. Anti-MFF (HPA010968, Sigma) and anti-MFF

(Proteintech, Chicago, IL, 17090–1-AP) were used at 1/50 and 1/2000, respectively, for immunofluorescence staining.

## Immunofluorescence staining

Cells were fixed by directly adding 16% paraformaldehyde into the growth medium to dilute to 4%. The reaction was quenched using PBS/Glycine 100 mM. Cells were then permeabilized using PBS/ 0.1% Trion X-100, and stained using the respective primary antibodies and goat anti-Rabbit secondary antibody conjugated to Alexa-568 diluted in 5% BSA.

## CRISPR knockout of *DRP1* gene

In order to generate a CRISPR-mediated knockout of the DRP1 locus we co-transfected KERMIT cells (*Kanfer et al., 2015*) in 10 cm dishes with 16.2 μg of pX330-DRP1ex2, 16.2 μg of pX330-DRP1ex6, and 3.6 μg of HcRed plasmid (*Kanfer et al., 2015*) for selecting transfected cells. Cell sorting was performed 48 hr post transfection (h.p.t.) and cells were seeded at very low density in 10 cm cell culture dishes. Colonies were individually transferred into 96-well plates, genotyped and analyzed by western blotting for DRP1 expression.

## Lentivirus production and transduction

Lentiviruses were produced in HEK293FT cells using the pPAX2 and pMD2.G packaging plasmids, kindly provided by Michael Way (London). HEK293FT cells were transfected with the packaging and transfer plasmids using FuGENE6 (Promega, Madison, WI) (7 μg pPAX2, 3 μg pMD2.G, 10 μg transfer plasmid complexed with 51 μl FuGENE6 in a 10 cm dish). Fresh medium containing 10 mM HEPES was provided 6–8 h.p.t.. Viruses were harvested in the next two days, pooled, filtered through 0.45 μm filters, aliquoted and frozen at −80°C for long-term storage.

For transduction experiments, viruses were diluted 1:16 in growth medium. U2OS cells were incubated in diluted virus overnight at 37°C. Fresh medium (containing 1 μg/ml puromycin if applicable) was provided the next day. Cells under selection were maintained in the presence of antibiotic for at least four days before being expanded and used for subsequent experiments.

## siRNA transfections

In order to knock-down DRP1 expression, small interfering RNAs targeting the gene in the coding region were obtained from Microsynth AG (Balgach, CH) [siRNA DRP1 #1, sense strand: 5′-UCCG UGAUGAGUAUGCUUUdTdT-3′ (*Koch et al., 2004*). Another sequence (siRNA DRP1 #2: 5′-CTGGA-GAGGAATGCTGAAA-3′ [*Wang et al., 2011*]) was also used but was less effective in our hands (*Figure 1—figure supplement 1A*). $0.075 \times 10^6$ cells were seeded 24 hr before transfection in 6-well plates (for fixing as well as for protein extracts). 40 nM of siRNA were transfected three times within 72 hr (24, 48 and 72 hr) using 2 μl Lipofectamine RNAiMax (Invitrogen, Carlsbad, CA). For MFF knockdown [siRNA, sense strand: 5′-CUGAGCAGUUCUGCAGUAACAdTdT-3′] 10 nM siRNA was transfected and cells were analyzed 48 hr later. siRNA against INF2-CAAX [5′- ACAAAGAAACTGTG TGTGA-3′] has been previously described (*Korobova et al., 2013*).

## *Shigella* infection and imaging

*Shigella flexneri* serotype 5a strain M90T stably expressing GFP or RFP were kindly provided by Jost Enninga (Paris). mCherry-tagged *Shigella* was described previously (*Mostowy et al., 2013*). *Shigella* were cultured overnight in Tryptic soy (TCS), diluted 50x in fresh TCS, and cultured until $OD_{600nm}$ = 0.6 as previously described (*Mazon Moya et al., 2014*). Virulent colonies were selected on Congo Red-containing TCS plates. Positive colonies were cultured in TCS medium.

For mCherry-Lifeact or DRP1 overexpression, plasmids were transfected using jetPEI (Polyplus, Illkirch, FR). Briefly, 1 μg plasmid and 4 μl jetPEI were separately diluted in 150 μl 150 mM NaCl and incubated at room temperature (RT) for 5 min. These reagents were then mixed and incubated at RT for 30 min before being added to cells in 6-well plates containing 2 ml growth medium. For fluorescently-tagged DRP1 transfection, DRP1-expressing plasmids were mixed with a pcDNA3 empty vector at approximately 1:10 ratio in order to obtain low expression level.

On the day of infection (24 h.p.t. when applicable), overnight *Shigella* cultures were diluted 50x and returned to the shaker at 37°C for 2.5 hr. 200 μl of the culture was then added in serum-free,

antibiotic-free medium, to cells in 6-well culture plates and spinoculation was performed at 100 x g for 10 min at room temperature. The plate was returned to the 37°C incubator for 30 min. Inoculum was then removed and cells were provided with fresh growth medium containing 50 μg / ml gentamycin to eliminate extracellular bacteria. *Shigella*-infected cells were imaged on either a Spinning Disk microscope (Nikon Eclipse T1) equipped with a Yokogawa Confocal Scanner Unit CSU-W1-T2, using a 100 × 1.49 CFI Apo TIRF oil objective and a sCMOS camera, or using a confocal microscope LSM 710 (Carl Zeiss MicroImaging) operated by ZEN 2010 software.

## AFM experiments

We used a Nanowizard I AFM atomic force microscope (JPK Instruments, Berlin, DE) with a SD-sphere-CONT-M-10 colloidal tip (sphere diameter of 2 μm, nominal stiffness of ~2 N/m, NANOSENSORS, Neuchatel, CH) treated for 60 s in oxygen plasma, coated with PLL-g-PEG as anti-fouling layer, and calibrated using the Sader method (*Sader et al., 1999*). The AFM was mounted on the stage of a laser scanning microscope (LSM 510, Carl Zeiss AG) for optical imaging. Cells were incubated at 37°C during all experiments.

## Cell culture on vinyl disks

Small disks (18 mm diameter) were cut out of second-hand gramophone records (*Franklin, 1986*; *Loggins, 1982*; *Raven, 1985*), disinfected with 70% ethanol, and transferred to 6-well dishes for cell culture and siRNA transfections. For quantification of the knock-down experiments, cells were fixed with 4% paraformaldehyde and the disks were mounted directly on a coverslip in 20 μl VectaShield mounting medium (Vectorlabs, Burlingame, CA) for imaging.

For GFP-MFF imaging, a thin layer of PDMS was spin-coated with a WS-650MZ-23NPP spincoater (3500 rpm, 15 s, Laurell Technologies Corporation, North Wales, PA) onto disks of approximately 6 cm in diameter cut out of the gramophone records. The disks were further cut to a diameter of ~3 cm and the PDMS layer was removed in the center, leaving only PDMS spacers on the outer edges of the disks. Cells depleted of DRP1 via lentivirus transduction were seeded on the disk and transduced with pLVX-Puro-MFF. After 24 hr, vinyl disks were flipped over and transferred to a glass-bottom 6-well plate, such that the disk rested on the PDMS spacers with cell-seeded side facing down. A two-Swiss-franc coin was used as weight on top of the vinyl disk in order to keep it close to the glass surface. The same setup was used for imaging of CLIMP-63 and ATL1-K80A transfected cells. U2OS cells stably expressing mtBFP and Sec61-GFP were seeded on the disk, and transfected 24 hr later with 600 ng of either CLIMP-63 or ATL1-K80A, 300 ng cytoplasmic RFP as transfection control, and 2 μg of empty vector (pBluescript KS (+), complexed with 11.8 μl of jetPEI reagent. Imaging took place 48 hr after transfection. During time-lapse experiments cells were incubated in DMEM without phenol red at 30°C, 5% $CO_2$.

Live Imaging was performed either with a water-dipping objective (40x W, 0.8 NA Leica HCX APO) on a Leica Upright microscope (DM6000B), or with a 60 × 1.4 CFI Plan Apo λ Oil on a Yokogawa CSU-W1-T2 spinning disk unit mounted on a Nikon Eclipse T1 microscope.

## Image processing

Images processing was performed using Fiji ImageJ. When applicable, images were background subtracted, denoised using the ndsafir program (*Boulanger et al., 2010*), and Bleach Correction was applied when needed, in the Simple Ratio mode.

## Statistics

For *Shigella* experiments, 23 constriction events were observed in wild-type conditions, which resulted in 13 fission events (from three independent experiments). 50 constrictions were observed in DRP1[CRISPR] cells, which resulted in 0 fission event (from two independent experiments). 19 constriction events were observed in DRP1-siRNA-treated cells, which resulted in 0 fission event (from three independent experiments). 13 constriction events were observed in MFF-siRNA-treated cells, which resulted in one fission event. For unstimulated mitochondria, 19 mitochondria were followed during 10–150 min. For AFM experiments, a total of 18 touchdowns led to a visible reduction of mitochondrial matrix stain, 16 of which led to fission (in three independent experiments). In DRP1 siRNA-treated cells, 32 successful touchdowns led to eight fission events (in three independent

experiments). In CytD-treated cells, 17 touchdowns led to 16 fission events (in two independent experiments). For vinyl records experiments, 84 scrambled-siRNA-treated, 72 DRP1-siRNA-treated cells, 120 MFF-siRNA-treated cells, 38 ATL1-K80A overexpressing cells and 21 CLIMP63 overexpressing cells were counted, of which 67, 10, 46, 34 and 18 had a divided mitochondrial network, respectively (from 3 to 5 independent experiments). 90 wild-type and 88 DRP1$^{CRISPR}$ cells were counted, in which 77 and 36 had a divided mitochondrial network, respectively (from three independent experiments). All statistical tests were computed using the Fisher's exact test in MatLab.

## Monte Carlo (MC) computer simulations

A hollow membrane tube was simulated as a triangulated network of beads where flipping is allowed between bonded beads to model membrane fluidity (*Šarić and Cacciuto, 2012*). Proteins were modelled as a linear chain of five spheres positioned at a circular arc with a radius of $R_{pr}$. The diameter of each sphere was σ (≈20 nm according to the biophysical parameters used herein). The center-to-center distance between the spheres within a protein was adjusted to 2/3 of the diameter of the individual spheres. The two 'wings' of the protein (constituted by the two terminal spheres) were rigid and allowed to rotate around the middle sphere. We also incorporated the flexibility between the wings by imposing an angular potential between them as: $U_{flex} = \frac{1}{2}k_{flex}(\theta - \theta_{eq})$, where $k_{flex}$ and $\theta_{eq}$ are the strength of the potential and the equilibrium angle between the two wings, respectively. We set these parameters as $k_{flex} = 20\,kT$ and $\theta_{eq} = 2\pi/3$. In addition, the five spheres were allowed to rotate, as a whole, around a random axis, changing the angle between protein pairs.

Both lipids and proteins were modelled using a coarse-grained approach, extensively utilized to study membrane associated phenomena (*Noguchi, 2009*; *Saric and Cacciuto, 2013*). The classical Canham-Helfrich model (*Helfrich, 1973*) (in the form of a dihedral angle potential) was used to define the curvature energy ($U_{CH}$) of the membrane:

$$U_{CH} = \kappa \sum_{<ij>} 1 - \boldsymbol{n}_i \cdot \boldsymbol{n}_j,$$

where $\boldsymbol{n}_i$ and $\boldsymbol{n}_j$ are the normal vectors of any pair of triangles and κ is the bending modulus of the membrane. The summation runs over all neighboring pairs of triangles $i$ and $j$. The energy cost associated with area changes of the membrane is included as:

$$U_{dA} = \gamma\,dA,$$

where γ and $dA$ are the surface tension and the change in the surface area, respectively. No conditions were imposed on the total area and volume of the tube. To allow the proteins to adhere to the membrane, a ligand-receptor-like attraction between the spheres and membrane beads was introduced, given by:

$$U_{adh} = -\epsilon\left(\frac{D_{min}}{r}\right)^6,$$

where ε is the adhesive strength, $D_{min}$ is the sum of sphere and membrane bead radii, and $r$ is the center-to-center distance between a sphere and membrane beads, respectively. Parameters were set to the biologically relevant values of: κ = 20 $k_bT$, ε = 3.8 $k_bT$, $D_{col}$ = 4 σ and γ = 1 $k_bT/\sigma^2$ (*Derényi et al., 2002*), where $\sigma$ is the diameter of a membrane bead and represents the length-scale in our simulations. In our model, $\sigma$ is approximately equivalent to 20 nm, hence spheres that construct the protein were set to have a diameter of 80 nm and the membrane surface tension is γ = 0.01 pN/nm. The total energy of the system is given by $U_{Tot} = U_{adh} + U_{dA} + U_{CH}$. For the special case of the pre-constricted tubes, we add a harmonic potential, given by

$$U_{CP} = \frac{1}{2}k_{CP}\left(R - R_{eq}\right)^2$$

where $k_{CP}$ and $R_{eq}$ are the strength of the constriction potential and the equilibrium constriction radius, respectively. We set these parameters as $k_{CP} = 0.01\,kT$ and $R_{eq} = 3\,\sigma$. MC simulations were performed in the npT ensemble with periodic boundary conditions, with p=0, to generate the most

energetically favorable configuration via the common Metropolis algorithm (*Metropolis et al., 1953*).

## Acknowledgements

We wish to thank Jasmine Abella and Michael Way (The Francis Crick Institute, London) for kindly sharing the lentivirus production system and the GFP/RFP-Lifeact constructs, Jost Enninga (Pasteur, Paris) for kindly providing the GFP/RFP-tagged *Shigella* strain, and Robin Klemm, Sumit Pawar and Ulrike Kutay for plasmids. BK was supported by the ERC (337906-OrgaNet) and an SNF-professor-ship (PP00P3_133651), SCJH by the ETH (ETH-16 13–2), QF by a EMBO fellowship (ALTF 1115–2014) and a fellowship from the Netherlands Organisation for Scientific Research (NWO, Rubicon 825.14.023), AV by a grant from NWO/OCW as part of the Frontiers of Nanoscience program, SM by a Wellcome Trust Research Career Development Fellowship (WT097411MA) and the Lister Institute of Preventive Medicine, and AŠ by the Royal Society University Research Fellowship. Microscopy was performed in part at the Scientific Center for Optical and Electron Microscopy of the ETH Zurich.

## Additional information

### Funding

| Funder | Grant reference number | Author |
| --- | --- | --- |
| European Commission | 337906-Organet | Sebastian Carsten Johannes Helle<br>Qian Feng<br>Benoît Kornmann |
| Eidgenössische Technische Hochschule Zürich | ETH-16 13-2 | Sebastian Carsten Johannes Helle<br>Benoît Kornmann |
| European Molecular Biology Organization | ALTF 1115-2014 | Qian Feng |
| Nederlandse Organisatie voor Wetenschappelijk Onderzoek | Rubicon 825.14.023 | Qian Feng |
| Nederlandse Organisatie voor Wetenschappelijk Onderzoek | OCW | Afshin Vahid Belarghou |
| Lister Institute of Preventive Medicine | | Serge Mostowy |
| Wellcome | WT097411MA | Serge Mostowy |
| Royal Society | | Anđela Šarić |
| Schweizerischer Nationalfonds zur Förderung der Wissenschaftlichen Forschung | PP00P3_13365 | Benoît Kornmann |

The funders had no role in study design, data collection and interpretation, or the decision to submit the work for publication.

### Author contributions

Sebastian Carsten Johannes Helle, Conceptualization, Resources, Formal analysis, Investigation, Methodology, Writing—original draft; Qian Feng, Conceptualization, Resources, Formal analysis, Funding acquisition, Investigation, Methodology, Writing—original draft; Mathias J Aebersold, Luca Hirt, Raphael R Grüter, Investigation, Methodology, AFM experiments; Afshin Vahid, Software, Formal analysis, Investigation, Visualization, Methodology, Writing—original draft, modeling; Andrea Sirianni, Investigation, Methodology, Training QF for shigella experiments; Serge Mostowy, Resources, Supervision, Writing—review and editing, Training QF for shigella experiments; Jess G Snedeker, Supervision, Funding acquisition, Methodology, AFM experiments; Anđela Šarić, Resources; Timon Idema, Software, Methodology, Writing—review and editing, Modeling; Tomaso Zambelli, Software, Supervision, Funding acquisition, Methodology, Writing—review and editing, modeling;

Benoît Kornmann, Conceptualization, Formal analysis, Supervision, Funding acquisition, Methodology, Writing—original draft, Project administration

### Author ORCIDs
Afshin Vahid, https://orcid.org/0000-0001-8540-3092
Anđela Šarić, https://orcid.org/0000-0002-7854-2139
Timon Idema, http://orcid.org/0000-0002-8901-5342
Benoît Kornmann, https://orcid.org/0000-0002-6030-8555

### Decision letter and Author response
Decision letter https://doi.org/10.7554/eLife.30292.035
Author response https://doi.org/10.7554/eLife.30292.036

---

## Additional files

### Supplementary files
• Transparent reporting form
DOI: https://doi.org/10.7554/eLife.30292.034

---

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
