## [Decision Letter]

[Editors’ note: a previous version of this study was rejected after peer review, but the authors submitted for reconsideration. The first decision letter after peer review is shown below.]

Thank you for submitting your work entitled "Mechano‐Induced Mitochondrial Fission" for consideration by *eLife*. Your article has been evaluated by Ivan Dikic (Senior Editor) and three reviewers, one of whom, Richard Youle, is a member of our Board of Reviewing Editors. The following individual involved in the review of your submission has agreed to reveal their identity: Mariusz Karbowski.

Our decision has been reached after consultation between the reviewers. Based on these discussions and the individual reviews below, we regret to inform you that your work will not be considered further for publication in *eLife*. Although all three reviewers find the results interesting and potentially important, they agreed that the study lacks sufficient exploration of mechanism and physiological context and is thus too preliminary considering the rapid revision cycle required for *eLife*.

*Reviewer #1:*

Helle and Feng et al. observe that mitochondria divide following physical perturbations; either by intracellular bacterium collision or external cues induced by atomic force microscopy and growth on vinyl records. The data in both the figures and movies provide convincing evidence that this phenomenon occurs. The authors suggest this mitigates network conflicts and perhaps other organelle entanglements. Although interesting, no mechanisms are revealed.

1) The authors offer no insight into how the mitochondrion senses these mechanical perturbations. It is interesting that Drp1 is recruited to areas where the bacterium touches the mitochondria or at the edge of an environmental shift in elevation. However, more mechanistic insight into Drp1 recruitment such as if and how MFF or Mid49/51 may respond to mechanical stress, stretching or mitochondrial constriction site generation would be important to expand the scope of the study. Does ER wrapping induce mechanical stain on mitochondria? Do ER contact sites participate in force generated mitochondrial fission?

2) The authors note that mechanically induced mitochondrial fission could have physiological implications but do not show any evidence for such. Finding examples of this would greatly increase the significance of the work.

*Reviewer #2:*

The authors show that mitochondrial fission can be triggered through mechanical forces including collision with intracellular pathogens (Figure 1), by external pressure applied by an atomic force microscope (Figure 2), or by cell migration across uneven surfaces (Figure 3). This paper suggests that the mitochondrial network is responsive to mechanical forces, which may equip the organelle to cope with intracellular crowdedness. However, the results are largely observational and the physiological rationale for the observed cellular phenomenon is not clear to me. The cytoplasm is full of hundreds of large trafficking organelles and if mechanical force was sufficient for fission, then mitochondrial fission would be excessive, but in reality it is infrequent in these cells.

*Reviewer #3:*

In the manuscript by Helle et al. the authors provide some evidence supporting the possibility that mechanical stimuli can lead to activation of Drp1-dependent mitochondrial fission.

Most of the conclusions are supported by high quality fluorescence imaging data. Furthermore, the crosstalk between mechanical force/subcellular crowdedness and mitochondrial dynamics has not been systematically studied. Thus these findings are novel and potentially interesting for the broad readership of *eLife*.

However, the link between Drp1-mediated mitochondrial fission and movement of the mitochondria from mechanically stressed areas is not supported in sufficient depth. The physiological role for this process and crosstalk with other mitochondria shaping processes is also less well developed (see below).

1) The findings described in this work are consistent with already published role of ER tubules in mitochondrial division (e.g., movement of ER tubules into the prospective mitochondrial fission sites). However, as suggested by others the ER tubules could serve as a specific part of the mitochondrial fission process, or perhaps (considering the authors conclusions) they could force mitochondrial fission in a less specific manner (a "crowd" model). Thus it would be interesting if the authors performed analyses of ER redistribution at least during Shigella-induced mitochondrial fission. Do ER tubules move together with fluorescent *Shigella* to facilitate mitochondrial fission?

2) In some movies mitochondrial fragmentation occurs relatively late after the mechanical stress is applied (e.g. after fluorescent *Shigella* moved away from affected area). This indicates that the conclusion that mitochondrial fission serves as a means to make space for other subcellular compartments may not be correct. Could the authors address this issue in more detail? What would be the alternative role of mechanical stress-induced mitochondrial fission?

3) While the imaging data shown in this work are of a high quality the quantitative assessment of Drp1 recruitment during force induced mitochondrial fission is not shown. Considering the possibility that mitochondria in Drp1 depleted cells do not redistribute because they are highly interconnected, while control cell mitochondria are already shorter and less interconnected and therefore could redistribute from the stressed areas even without activation of mitochondrial fission, a more quantitative assay on Drp1 foci formation and mitochondrial fission would be necessary to further substantiate this important point. For example the authors could quantify Drp1 foci-mediated fission in stressed and unstressed areas. Considering resolution of fluorescent imaging without such studies it is difficult to assess whether indeed mitochondrial fission per se or movement of preexisting mitochondria is affected by the mechanical stress.

[Editors’ note: what now follows is the decision letter after the authors submitted for further consideration.]

Thank you for submitting your article "Mechanical Force Induces Mitochondrial Fission" for consideration by *eLife*. Your article has been favorably evaluated by Ivan Dikic (Senior Editor) and three reviewers, one of whom is a member of our Board of Reviewing Editors. The reviewers have opted to remain anonymous.

The reviewers have discussed the reviews with one another and the Reviewing Editor has drafted this decision to help you prepare a revised submission.

Summary:

The manuscript, "Mechano-Induced Mitochondrial Fission" reports that mitochondrial fission can be triggered by mechanical forces caused by both intracellular (pathogens) and extracellular (cell migration across uneven surfaces) forces. This paper is interesting in that it details a mechanism by which the mitochondrial network can respond to mechanical forces thereby equipping the organelle with the ability to cope with intracellular crowdedness. Using a variety of mechanical-inducing mechanisms the authors convincingly show that mechanical force is sufficient to induce Drp1-dependent mitochondrial fragmentation.

Essential revisions:

1) The role of actin polymerization on the OMM should be investigated in more depth. The application of cytochalasin D disrupts cell structure and perturbs several cellular events. While nice approaches were used here to eliminate or reduce Drp1 expression, experiments on the role of actin in force-induced mitochondrial fission are underdeveloped. Does INF2- or Spire-depletion affect this process? It is also surprising that the authors did not show mitochondria in cells shown in Figure 4.

2) The statement (subsection “Mechanism of force sensing and DRP1 recruitment”, first paragraph) that the adaptor proteins (e.g. Mff or MiD49/51) are recruited to presumptive fission sites upstream of Drp1 is not correct, especially in case of Mff. Drp1 association with the mitochondria induces Mff foci on the OMM (therefore, endogenous Mff foci can be easily detected in wild type cells, but are missing in Drp1^-/-^ cells, where Mff shows diffuse localization). Thus, while the role proposed here for Mff in sensing mechanical stress of the mitochondria is very interesting, the authors should provide some evidence that in the absence of Drp1, endogenous Mff indeed accumulates on highly constricted/tubulated mitochondria. The authors could perform immunofluorescence of Drp1^-/-^ cells growing on vinyl discs etc. Without such data, I have a problem believing that these GFP-Mff-based experiments reflect function of endogenous Mff in the process.

3) The data presented in Figure 4 does not seem to support the authors' conclusion that the "ER is not necessary for mechanical force induced fission". While the ER does appear to be "pushed away" following AFM stimulation there appears to be a tubule (panel A) that localizes to the same vicinity as the mitochondrial constriction prior to fission. However, it is nearly impossible to really resolve what is going on at the ER level with the tiny insets, and resolution of the images (ER staining is unclear as to what is really ER structure in panels A, B and C). Merged images in dual colors, as well as better resolution of the ER network will help to clarify whether the ER is recruited or not. The authors may need to look for fission events in cells with resolvable ER tubules to address whether the ER has a role or not.

4) Do the authors see any change in the rate of mechanical force induced mitochondrial fission vs. non-mechanical force induced following defects in ER shape (+Climp63) or tubule fusion (+ATL1-K80A)? The authors comment that they still visualized fission in both conditions, however, there is no mention to how frequent or what the sample size was. Quantification of the imaging observed in Figure 4 as to the rate and frequency of these events would strongly strengthen the conclusions from this figure. In addition, others have found that even when tubular ER is reduced (as in Reticulon depleted cells), mitochondrial fission still occurs at the remaining ER tubules still present. And even when Climp63 is overexpressed there are a lot of tubules still around. As it is impossible to fully deplete cells of ER, the strong conclusion about the role of ER in mechanical force-induced mitochondrial fission should be tempered.

5) The authors should comment on whether they would consider ER-mediated constriction as a form of mechanical force.

6) In reference to Figure 4, the authors report that mechanically induced fission still took place with rates similar to untreated cells. Can the authors expand on what the rate was and add quantification to Figure 4 to demonstrate this conclusion?

---

## [Author Response]

[Editors’ note: the author responses to the first round of peer review follow.]

Reviewer #1:Helle and Feng et al. observe that mitochondria divide following physical perturbations; either by intracellular bacterium collision or external cues induced by atomic force microscopy and growth on vinyl records. The data in both the figures and movies provide convincing evidence that this phenomenon occurs. The authors suggest this mitigates network conflicts and perhaps other organelle entanglements. Although interesting, no mechanisms are revealed.

This reviewer was rather satisfied with the quality of the data and our conclusion that force-induced mitochondrial fission occurs. S/he also found the implication that this phenomenon may reduce organelle entanglement interesting. We thank this reviewer for this positive feedback. We think that we have addressed the lack of mechanism (see below).

1) The authors offer no insight into how the mitochondrion senses these mechanical perturbations. It is interesting that Drp1 is recruited to areas where the bacterium touches the mitochondria or at the edge of an environmental shift in elevation. However, more mechanistic insight into Drp1 recruitment such as if and how MFF or Mid49/51 may respond to mechanical stress, stretching or mitochondrial constriction site generation would be important to expand the scope of the study. Does ER wrapping induce mechanical stain on mitochondria? Do ER contact sites participate in force generated mitochondrial fission?

We have investigated the mechanism of force-sensing. In this version of the manuscript, we show that MFF has a preference for mitochondrial tubules with a reduced diameter. Mechanical stimulation of mitochondria induced local constriction of the mitochondrial tubules and enrichment of MFF at these sites (Figure 5). We further show that MFF, when overexpressed to a high level, was able to induce and stabilize thin mitochondrial tubules in the absence of exogenous mechanical stimulation (Figure 5). This apparent paradoxical behavior of MFF could be modeled by a computer-assisted simulation of a minimal system, which shows that the ability to sense and induce constrictions are coupled (see Figure 6, Figure 6—figure supplement 1, Video 17–Video 19).

We also addressed the role of ER and actin in mechano-fission. We show in the current version that mechanically induced mitochondrial fission can still take place under conditions where ER dynamics is disturbed (i.e. upon overexpression of dominant-negative atlastin or wildtype CLIMP-63) or where actin polymerization is inhibited (i.e. treatment of cytochalasin D). These data are presented in Figure 4 and Video 10–Video 14). We now explicitly express the potential roles of ER and actin as force-inducer in the Discussion section (third paragraph).

2) The authors note that mechanically induced mitochondrial fission could have physiological implications but do not show any evidence for such. Finding examples of this would greatly increase the significance of the work.

While we obviously agree with this reviewer that evidence of physiological function of mechanically induced fission would greatly increase the significance of our study, we have to point out that the physiological roles of ER wrapping-induced mitochondrial fission, and in fact, fusion and fission in general, have also largely remained speculative.

In the case of mechanically induced mitochondrial fission, we believe that there could be two conceivable physiological functions. Firstly, using serial block-face scanning electron microscopy we have observed concatenations (locked rings) between ER and mitochondria in both untreated and DRP1 shRNA-treated cells (see Author response image 1). Since ER fission is not frequently observed, these entanglements presumably have to be resolved by mitochondrial fission. Similar entanglements and local clashes may also occur between mitochondria and other cellular structures. Secondly, it is long known that immune and cancer cells are capable of extreme forms of deformation during migration (Proebstl et al., 2012; Starke et al., 2013). Two recent papers published back-to-back in Science (Denais et al., 2016; Raab et al., 2016) showed that cells “squeeze” to such an extent that the nuclear membrane transiently loses integrity. Under such conditions, we assume that mitochondria, and other organelles, must also adapt, and it is tempting to imagine that mechano-sensing by MFF facilitates controlled fission at the bottle-neck, resolving internal physical tension during such deformations.

We, however, believe that these questions are beyond the scope of this current manuscript, which is aimed at describing the novel phenomenon that a mechanical trigger can be sensed by mitochondria and translated to a biochemical response, namely the recruitment of the fission machinery.

**Author response image 1. respfig1:** Concatenations between the ER and mitochondria in the absence of mitochondrial fission machinery. Two examples of concatenations observed in DRP1 shRNA-treated U2OS KERMIT cells. The relevant parts of the ER (green) and mitochondria (red) were traced manually and 3D reconstituted with Fiji. These representations do not faithfully represent the morphology of the organelles, but show their relative topology, highlighting concatenations.

Reviewer #2:The authors show that mitochondrial fission can be triggered through mechanical forces including collision with intracellular pathogens (Figure 1), by external pressure applied by an atomic force microscope (Figure 2), or by cell migration across uneven surfaces (Figure 3). This paper suggests that the mitochondrial network is responsive to mechanical forces, which may equip the organelle to cope with intracellular crowdedness. However, the results are largely observational and the physiological rationale for the observed cellular phenomenon is not clear to me. The cytoplasm is full of hundreds of large trafficking organelles and if mechanical force was sufficient for fission, then mitochondrial fission would be excessive, but in reality it is infrequent in these cells.

This reviewer assumes that mitochondria are constantly under mechanical strain from collisions with other structures. As a matter of fact, this intuition is not verified by our observations. We have a very efficient proxy for mechanical strain, since the matrix stain is systematically “squeezed” away from the zone of mechanical stimulation. This proxy indicates that in resting cells, mitochondria are only seldom constricted. The infrequent observation of spontaneous constriction actually agrees with the relatively infrequent observation of fission events.

We would like to refer to our response to reviewer #1 point 2 concerning the physiological rationale of mechano-induced mitochondrial fission.

Reviewer #3:In the manuscript by Helle et al. the authors provide some evidence supporting the possibility that mechanical stimuli can lead to activation of Drp1-dependent mitochondrial fission.Most of the conclusions are supported by high quality fluorescence imaging data. Furthermore, the crosstalk between mechanical force/subcellular crowdedness and mitochondrial dynamics has not been systematically studied. Thus these findings are novel and potentially interesting for the broad readership of eLife.However, the link between Drp1-mediated mitochondrial fission and movement of the mitochondria from mechanically stressed areas is not supported in sufficient depth. The physiological role for this process and crosstalk with other mitochondria shaping processes is also less well developed (see below).

This reviewer was very positive about the quality of our data, the conclusions we drew, as well as the novelty of the finding. We thank this reviewer for this positive feedback.

S/he also raised the point that the mechanism of how physical force is translated to Drp1 recruitment is not sought out in depth, and that the physiological role is not well developed. We think that our new data and models concerning force sensing by MFF adequately address this point.

1) The findings described in this work are consistent with already published role of ER tubules in mitochondrial division (e.g., movement of ER tubules into the prospective mitochondrial fission sites). However, as suggested by others the ER tubules could serve as a specific part of the mitochondrial fission process, or perhaps (considering the authors conclusions) they could force mitochondrial fission in a less specific manner (a "crowd" model). Thus it would be interesting if the authors performed analyses of ER redistribution at least during Shigella-induced mitochondrial fission. Do ER tubules move together with fluorescent Shigella to facilitate mitochondrial fission?

We would like to refer to our response to reviewer #1 point #1 3^rd^ paragraph.

2) In some movies mitochondrial fragmentation occurs relatively late after the mechanical stress is applied (e.g. after fluorescent Shigella moved away from affected area). This indicates that the conclusion that mitochondrial fission serves as a means to make space for other subcellular compartments may not be correct. Could the authors address this issue in more detail? What would be the alternative role of mechanical stress-induced mitochondrial fission?

We also noticed that the time elapsed between *Shigella* impact and eventual fission varied. Technically, we think that it is likely that Drp1 needs to be recruited, thus requiring some time before fission is executed.

However, we also think that force-induced mitochondrial fission did not evolve to cope with *Shigella* infection, but rather to cope with much slower stimulations from intracellular origin. Actually, most of the time, *Shigella* does not represent a threat for mitochondria, as it can swim above or below, only leading to a very transient constriction (Figure 1).

We think that mechano-induced mitochondrial fission may not only serve to make space for other subcellular compartments, but also to avoid entanglement of the mitochondrial network itself. Concatenations such as those we show above may result in force being applied on mitochondria for extended amounts of time.

3) While the imaging data shown in this work are of a high quality the quantitative assessment of Drp1 recruitment during force induced mitochondrial fission is not shown. Considering the possibility that mitochondria in Drp1 depleted cells do not redistribute because they are highly interconnected, while control cell mitochondria are already shorter and less interconnected and therefore could redistribute from the stressed areas even without activation of mitochondrial fission, a more quantitative assay on Drp1 foci formation and mitochondrial fission would be necessary to further substantiate this important point. For example the authors could quantify Drp1 foci-mediated fission in stressed and unstressed areas. Considering resolution of fluorescent imaging without such studies it is difficult to assess whether indeed mitochondrial fission per se or movement of preexisting mitochondria is affected by the mechanical stress.

This reviewer raised a good point, and we have done exactly as suggested. We found that quantifying Drp1 foci was not very informative because a majority of these foci were actually not linked to a fission event. By contrast we could observe DRP1 foci on most fission sites. Thus we decided to quantify fission events instead.

Individual, identifiable mitochondria in the edge and ledge areas on vinyl disks were followed for as long as they remained in focus, or until the end of the movie acquisition. The amount of time elapsed before a mitochondrion underwent fission is plotted in Author response image 2 (Red circles). For mitochondria that did not undergo fission within observation period, we plotted the total amount of time the mitochondria were under observation (Black circles). Note that the maximal plottable time is 1000 seconds, the total length of the acquisition sessions. We also added these data in the Results section as following: “Indeed, a few minutes into time-lapse recording, mitochondria vacated the edge area by first undergoing fission (averagely within 93 sec), and then moving towards the ledge and the groove areas (Figure 3 and Video 1). The majority of mitochondria on the ledge area did not undergo fission within an average of 765 seconds (time in which they remained in focus, or until the end of the microscopy session).”

**Author response image 2. respfig2:** 

[Editors' note: the author responses to the re-review follow.]

Essential revisions:1) The role of actin polymerization on the OMM should be investigated in more depth. The application of cytochalasin D disrupts cell structure and perturbs several cellular events. While nice approaches were used here to eliminate or reduce Drp1 expression, experiments on the role of actin in force-induced mitochondrial fission are underdeveloped. Does INF2- or Spire-depletion affect this process? It is also surprising that the authors did not show mitochondria in cells shown in Figure 4.

Cytochalasin D is indeed toxic and has pleiotropic effects on cells. Our naive reasoning was that if actin polymerization had any role in force-induced mitochondrial fission, then hitting actin the most brutal way possible should show some effect on this process. However, cytochalasin D treatment did not affect mitochondria’s ability to sense or respond to mechanical forces, while it indeed disrupted cell structure, as shown in Figure 4, and undoubtedly perturbed several cellular events.

While these data strongly suggested that actin polymerization, as a whole, did not play measurable roles in mechano-fission, we nevertheless repeated the vinyl records and *Shigella* experiments in INF2-depleted cells. INF2 was previously shown to participate in ER wrapping-induced mitochondrial fission, and as the reviewer suggested, may play a more specific/subtle role in mechano-fission. We observed that the frequency of mechano-fission remained the same in INF2-depleted cells as in control cells in both experimental systems. These data have been added to the Results section of the manuscript, under the subtitle “Inhibiting ER or actin dynamics does not detectably affect force-induced fission” (last paragraph).

We have also attempted to examine the effect of Spire1C depletion in similar experiments. Spire1C is an alternate splice isoform of Spire1, an actin-nucleating factor. This specific isoform localizes to mitochondria where it interacts with INF2 and thereby facilitates ER tubule-mediated fission (Manor et al., 2015). We have tried to deplete Spire1C using two siRNA sequences derived from the two reported shRNA constructs (Manor et al., 2015). Although some knockdown effect was maybe attained, it was however clear that residual Spire1C signal, as immunostained using the Exon C antibody, also used in the original study, was found on the mitochondria (Author response image 3). We have contacted the authors of the Manor et al. study for the reported shRNA constructs but they could not send it to us for reasons beyond our control. We feel that we cannot make a decisive point on the involvement of Spire1C in the present manuscript, given the poor knock down effect. We feel, however, that the main conclusions of our study would not change whether or not Spire1C influences mechano-fission. Also, having performed the INF2 knockdown as well as cytochalasin D experiments, we believe we have sufficiently ruled out the INF2-specific pathway and actin polymerization, in a more general way, as essential players in mechano-fission. Moreover, our conclusions on this aspect do not exclude a subtle role of Spire1C, INF2 and actin in mechano-fission (“These results suggest that the INF2-mediated pathway does not play essential roles in mechanically induced fission” and “Thus, actin polymerization does not appear to be a necessary step in force-induced fission.”).

**Author response image 3. respfig3:** Immunostaining of Spire1C in cells treated with scrambled or Spire1C-targeting siRNAs. U2OS cells expressing mitochondrial matrix-targeting BFP (Mito) were transfected with either scrambled (SCR) or two siRNAs targeting either the ORF or 3’ UTR of Spire1C. Cells were fixed 72 hours post transfection and immunofluorescence staining was performed using the Exon C antibody as previously described (Manor et al., 2015). Samples were then imaged using a 60x oil objective (1.4NA DIC Oil PlanApoN) on a DeltaVision microscope. All images were taken using the same settings and presented at the same intensity levels.

We have added the mitochondrial channel for the image in Figure 4, as suggested. Because this panel was simply meant to show that cytochalasin D treatment was effective at depolymerizing actin, we initially did not consider that adding the mitochondrial channel was informative.

Altogether, we think that showing that neither the subtle perturbation of specific actin effectors, nor the brutal disruption of actin dynamics by cytochalasin D had obvious effects on mitochondrial force sensing and response, consolidates our conclusions.

2) The statement (subsection “Mechanism of force sensing and DRP1 recruitment”, first paragraph) that the adaptor proteins (e.g. Mff or MiD49/51) are recruited to presumptive fission sites upstream of Drp1 is not correct, especially in case of Mff. Drp1 association with the mitochondria induces Mff foci on the OMM (therefore, endogenous Mff foci can be easily detected in wild type cells, but are missing in Drp1^-/-^ cells, where Mff shows diffuse localization). Thus, while the role proposed here for Mff in sensing mechanical stress of the mitochondria is very interesting, the authors should provide some evidence that in the absence of Drp1, endogenous Mff indeed accumulates on highly constricted/tubulated mitochondria. The authors could perform immunofluorescence of Drp1^-/-^ cells growing on vinyl discs etc. Without such data, I have a problem believing that these GFP-Mff-based experiments reflect function of endogenous Mff in the process.

First, we would like to apologize for forgetting to include a reference for our claim that “adaptor proteins are recruited to presumptive fission sites upstream of Drp1”. We were specifically thinking of the report from Friedman and colleagues, where MFF was shown to accumulate at ER tubule-driven constrictions in the absence of Drp1 (Friedman et al., 2011). Yet, Friedman et al. also used GFP-tagged MFF and not the endogenous one, therefore we guess that the reviewer’s criticism also applies there.

We thank the reviewers for pointing out a piece of literature that has apparently escaped our attention. We have however not found the said study(-ies) reporting that MFF shows a diffused staining pattern in Drp1^-/-^ cells in the literature. The only MFF immunofluorescence experiment performed in the absence of Drp1 we could find was from Otera and colleagues, using Drp1KD rather than KO cells (Otera et al., 2010) (Figure 2). Here is an excerpt of this paper:

“Immunofluorescence microscopy revealed that Mff is localized mostly in puncta on mitochondria. […] Interestingly, these structures were detected on the extended tubular networks in Drp1 RNAi cells but were not affected by hFis1 RNAi or Drp1/hFis1 double RNAi (unpublished data for Drp1/hFis1 RNAi), suggesting that Mff is present as preassembled structures on the MOM irrespective of Drp1 or hFis1 expression.”

Thus, it appears that the literature is controversial on this issue. Regardless, it is a great idea to examine endogenous MFF in our study, especially since good antibodies make this experiment completely feasible. We have observed endogenous MFF by immunofluorescence (IF) in control and DRP1-depleted cells and, in agreement with Otera et al., did not observe that “MFF localization shows diffuse localization” in either case (see below). Instead, endogenous MFF stained as punctae (Figure 5—figure supplement 1).

We have performed two additional sets of experiments with regard to endogenous MFF localization:

1) We performed IF of endogenous MFF in Drp1 KD cells, and observed that MFF foci show a preference for naturally occurring constrictions on mitochondria (Figure 5).

2) We developed a new protocol for immunofluorescence on vinyl records. Using this protocol, we show that endogenous MFF often accumulates at pattern-induced constrictions in Drp1-depleted cells (Figure 5—figure supplement 1).

Together these data nicely complement our GFP-tagging approach and demonstrate that the propensity of MFF to accumulate to constrictions is not an artifact of GFP tagging nor of overexpression. The new data on endogenous MFF have been added to the Results section of the manuscript, under the subtitle “Mechanism of force sensing and DRP1 recruitment” (first paragraph).

3) The data presented in Figure 4 does not seem to support the authors' conclusion that the "ER is not necessary for mechanical force induced fission". While the ER does appear to be "pushed away" following AFM stimulation there appears to be a tubule (panel A) that localizes to the same vicinity as the mitochondrial constriction prior to fission. However, it is nearly impossible to really resolve what is going on at the ER level with the tiny insets, and resolution of the images (ER staining is unclear as to what is really ER structure in panels A, B and C). Merged images in dual colors, as well as better resolution of the ER network will help to clarify whether the ER is recruited or not. The authors may need to look for fission events in cells with resolvable ER tubules to address whether the ER has a role or not.

We apologize for the image quality and resolution of the ER in panel A, B, and C. In fact, their quality is limited by constraints associated with either mounting an atomic-force microscope on an optical microscope, or with the imaging conditions on vinyl records (with very unusual refractive properties of the sample and at a considerable working distance from the microscope’s objective). Clearly resolving individual ER tubules is not possible with these setups.

We entirely agree with the reviewers’ opinion that the data from the AFM alone are not conclusive regarding the involvement of the ER, and apologize for the lack of clarity. We never meant to draw strong conclusions from the AFM experiment alone. Our original text read:

“When performing AFM stimulation of U2OS cells stably expressing both a mitochondrial (mtBFP) and an ER (GFP-Sec61β) fluorescent marker, we observed that the tip-mediated indentation of cells pushed the ER away from the site of force application and subsequent mitochondrial fission, suggesting that the ER was not necessary for force-mediated fission (Figure 4, Video 11). To investigate this further, we observed the ER together with mitochondria, in conditions where ER dynamics was perturbed by […]”

The sentence the reviewers referred to was not meant as a conclusion, but rather as an introduction to the set of experiments thereafter, involving perturbations of ER dynamics (CLIMP63 and ATL1DN overexpression). The actual conclusion for this set of experiment was:

“Thus, while the ER might have a function in force-induced fission, this function does not appear strictly necessary.”

This conclusion, we think, adopts an appropriately prudent tone and reflects what we observe: whatever we do to the ER, we do not see a detectable effect on force-induced fission. This conclusion is also nicely supported by the experiments in Figure 4 and Video 14, showing a clear event of force-induced fission happening at a considerable distance from any ER tubule. Nevertheless, to avoid any confusion, we have rephrased the introductory sentence to:

“However, when performing AFM stimulation of U2OS cells stably expressing both a mitochondrial (mtBFP) and an ER (GFP-Sec61β) fluorescent marker, we observed that the tip-mediated indentation of cells pushed the ER away from the site of force application and subsequent mitochondrial fission (Figure 4, Video 11). This did not appear consistent with an important role of the ER in force-induced fission. To investigate this further, we observed the ER together with mitochondria, in conditions where ER dynamics was perturbed by the overexpression of dominant negative Atlastin (ATL) mutants or of CLIMP-63.”

and removed the misleading part (“suggesting that […]”).

4) Do the authors see any change in the rate of mechanical force induced mitochondrial fission vs. non-mechanical force induced following defects in ER shape (+Climp63) or tubule fusion (+ATL1-K80A)? The authors comment that they still visualized fission in both conditions, however, there is no mention to how frequent or what the sample size was. Quantification of the imaging observed in Figure 4 as to the rate and frequency of these events would strongly strengthen the conclusions from this figure. In addition, others have found that even when tubular ER is reduced (as in Reticulon depleted cells), mitochondrial fission still occurs at the remaining ER tubules still present. And even when Climp63 is overexpressed there are a lot of tubules still around. As it is impossible to fully deplete cells of ER, the strong conclusion about the role of ER in mechanical force-induced mitochondrial fission should be tempered.

Following this suggestion, we have quantified the proportion of cells showing a complete separation of the mitochondrial network, when their cytoplasm is constricted by the record’s edge under the said conditions. We found this proportion to be 89% for ATL1-K80A cells (n=38 cells from 5 independent experiments) and 86% for CLIMP63 overexpressing cells (n=21 cells from 3 independent experiments). This is not significantly different from wildtype cells (79% and 85%, Figure 3). These data have been added to the manuscript (subsection “Inhibiting ER or actin dynamics does not detectably affect force-induced fission”, second paragraph and– subsection “Statistics”).

Hence, we cannot detect any obvious effect of ER disturbance on force-induced fission. This result is a negative one and should be interpreted as such. We wholeheartedly agree with the reviewer that it is impossible to exclude a contribution from the ER in force-induced fission at this stage, for the reasons that s/he explained above, and because this being a negative result, it is logically impossible to conclude. Indeed, “absence of evidence is no evidence for absence”. We however think that the conclusion that we draw is not strong (see response to point 3). We have additionally softened the header of the section from

“Inhibiting ER or actin dynamics does not affect force-induced fission”

to

“Inhibiting ER or actin dynamics does not detectably affect force-induced fission”

We would also like to point out that this set of experiments on ER involvement in force-induced fission was specifically made upon the request of reviewers in a first round of reviews for *eLife*. The only sensible conclusion that we can draw from these experiments is that whatever we do to the ER, we are not able to detect any notable effect on force-induced fission. We think that this is now better reflected in the text.

5) The authors should comment on whether they would consider ER-mediated constriction as a form of mechanical force.

We originally addressed this point in the Discussion with the following two sentences:

“It is tempting to speculate that the additional pressure imposed on the mitochondrial surface by crossing-over ER tubules, rigidified by an INF2- and Spire1- mediated actin shell, might be what triggers fission at ER-mitochondria encounter sites.”

and

“An appealing hypothesis is that the ER, being an intrinsic source of mechanical force, appears dispensable in our experimental conditions because an extrinsic source of mechanical force is present.”

We understand that these sentences might come across less affirmative than one might wish for, but these ideas are purely speculative at this stage, and to our opinion, should remain speculative. In attempt to state our hypotheses more directly, but without overstepping our confidence, we have rephrased them to:

“It is tempting to speculate that crossing-over ER tubules, rigidified by an INF2- and Spire1- mediated actin shell, might impose mechanical force to underlying mitochondria, triggering fission at ER-mitochondria encounter sites.”

and

“An appealing hypothesis is that the ER constitutes an intrinsic source of mechanical force. It therefore appears dispensable in our experimental conditions because an extrinsic source of mechanical force is present.”

6) In reference to Figure 4, the authors report that mechanically induced fission still took place with rates similar to untreated cells. Can the authors expand on what the rate was and add quantification to Figure 4 to demonstrate this conclusion?

We apologize for omitting the quantification of fission in cytochalasin D-treated cells in the “statistics” subsection of the Materials and methods section. This was an oversight, and these data should have been there from the beginning. We also apologize for the poor choice of the word “rates”. It should have rather been “proportion”.

Indeed, from 17 different AFM touchdowns, in two independent experiments, we observed 16 fission events, a proportion (94%) that is well in agreement with that of untreated cells (Figure 2). The quantification has been added to the manuscript (–subsection “Inhibiting ER or actin dynamics does not detectably affect force-induced fission”, last paragraph and subsection “Statistics”), and the word “rate”, replaced by “proportion”.